# Solving Linear-Gaussian Bayesian Inverse Problems with Decoupled Diffusion Sequential Monte Carlo

**Filip Ekström Kelvinius** [1]   **Zheng Zhao** [1]   **Fredrik Lindsten** [1]

## Abstract

A recent line of research has exploited pre-trained generative diffusion models as priors for solving Bayesian inverse problems. We contribute to this research direction by designing a sequential Monte Carlo method for linear-Gaussian inverse problems which builds on "decoupled diffusion", where the generative process is designed such that larger updates to the sample are possible. The method is asymptotically exact and we demonstrate the effectiveness of our Decoupled Diffusion Sequential Monte Carlo (DDSMC) algorithm on both synthetic as well as protein and image data. Further, we demonstrate how the approach can be extended to discrete data.

## 1. Introduction

Generative diffusion models (Sohl-Dickstein et al., 2015; Ho et al., 2020; Song et al., 2021b) have sparked an enormous interest from the research community, and shown impressive results on a wide variety of modeling task, ranging from image synthesis (Dhariwal & Nichol, 2021; Rombach et al., 2022; Saharia et al., 2022) and audio generation (Chen et al., 2020; Kong et al., 2021) to molecule and protein generation (Hoogeboom et al., 2022; Xu et al., 2022; Corso et al., 2023). A diffusion model consists of a neural network which implicitly, through a generative procedure, defines an approximation of the data distribution, $p_\theta(\mathbf{x})$. While methods (e.g. Ho & Salimans, 2022) where the model is explicitly trained to define a conditional distribution $p_\theta(\mathbf{x}|\mathbf{y})$ exist, this conditional training is not always possible. Alternatively, a domain-specific likelihood $p(\mathbf{y}|\mathbf{x})$ might be known, and in this case, using $p_\theta(\mathbf{x})$ as a prior in a Bayesian inverse problem setting, i.e., sampling from the posterior distribution $p_\theta(\mathbf{x}|\mathbf{y}) \propto p(\mathbf{y}|\mathbf{x})p_\theta(\mathbf{x})$, becomes an appealing alternative. This approach is flexible since many different likelihoods can be used with the same diffusion model prior, without requiring retraining or access to paired training data. Previous methods for posterior sampling with diffusion priors, while providing impressive results on tasks like image reconstruction (Kawar et al., 2022; Chung et al., 2023; Song et al., 2023), often rely on approximations and fail or perform poorly on simple tasks (Cardoso et al., 2024, and our Section 5.1), making it uncertain to what extent they can solve Bayesian inference problems in general.

Sequential Monte Carlo (SMC) (Del Moral, 2004; Naesseth et al., 2019) is a well-established method for Bayesian inference, and its use of sequences of distributions makes it a natural choice to combine with diffusion priors. It also offers asymptotical guarantees, and the combination of SMC and diffusion models has recently seen a spark of interest (Trippe et al., 2023; Dou & Song, 2023; Wu et al., 2023; Cardoso et al., 2024). Moreover, the design of an efficient SMC algorithm involves a high degree of flexibility while guaranteeing asymptotic exactness, which makes it an interesting framework for continued exploration.

As previous (non-SMC) works on posterior sampling has introduced different approximations, and SMC offers a flexible framework with asymptotic guarantees, we aim to further investigate the use of SMC for Bayesian inverse problems with diffusion priors. In particular, we target the case of linear-Gaussian likelihood models. We develop a method which we call Decoupled Diffusion SMC (DDSMC) utilizing and extending a previously introduced technique for posterior sampling based on decoupled diffusion (Zhang et al., 2024). With this approach, it is possible to make larger updates to the sample during the generative procedure, and it also opens up new ways of taking the conditioning on $\mathbf{y}$ into account in the design of the SMC proposal distribution. We show how this method can effectively perform posterior sampling on synthetic data, and its effectiveness on image and protein structure reconstruction tasks. We also further generalize DDSMC for discrete data. Code is available online[1].

[1]Department of Computer and Information Science (IDA), Linköping University, Sweden. Correspondence to: Filip Ekström Kelvinius <filip.ekstrom@liu.se>.

*Proceedings of the 42nd International Conference on Machine Learning*, Vancouver, Canada. PMLR 267, 2025. Copyright 2025 by the author(s).

---

[1]https://github.com/filipekstrm/ddsmc

## 2. Background

As mentioned in the introduction, we are interested in sampling from the posterior distribution $p_\theta(\mathbf{x}|\mathbf{y}) \propto p(\mathbf{y}|\mathbf{x})p_\theta(\mathbf{x})$, where the prior $p_\theta(\mathbf{x})$ is implicitly defined by a pre-trained diffusion model and the likelihood is linear-Gaussian, i.e., $p(\mathbf{y}|\mathbf{x}) = \mathcal{N}(\mathbf{y}|A\mathbf{x}, \sigma_y^2 I)$. We will use the notation $p_\theta(\cdot)$ for any distribution that is defined via the diffusion model, the notation $p(\cdot)$ for the likelihood (which is assumed to be known), and $q(\cdot)$ for any distribution related to the fixed *forward* diffusion process (see next section).

### 2.1. Diffusion Models

Diffusion models are based on transforming data, $\mathbf{x}_0$, to Gaussian noise by a Markovian *forward* process of the form[2]

$$q(\mathbf{x}_{t+1}|\mathbf{x}_t) = \mathcal{N}(\mathbf{x}_{t+1}|\mathbf{x}_t, \beta_{t+1}I). \tag{1}$$

The generative process then consists in trying to reverse this process, and is parametrized as a backward Markov process

$$p_\theta(\mathbf{x}_{0:T}) = p_\theta(\mathbf{x}_T) \prod_{t=0}^{T-1} p_\theta(\mathbf{x}_t|\mathbf{x}_{t+1}). \tag{2}$$

The reverse process $p_\theta$ is fitted to approximate the reversal of the forward process, i.e., $p_\theta(\mathbf{x}_t|\mathbf{x}_{t+1}) \approx q(\mathbf{x}_t|\mathbf{x}_{t+1})$, where the latter can be expressed as

$$q(\mathbf{x}_t|\mathbf{x}_{t+1}) = \int q(\mathbf{x}_t|\mathbf{x}_{t+1}, \mathbf{x}_0)q(\mathbf{x}_0|\mathbf{x}_{t+1})d\mathbf{x}_0. \tag{3}$$

While $q(\mathbf{x}_t|\mathbf{x}_{t+1}, \mathbf{x}_0)$ is available in closed form, $q(\mathbf{x}_0|\mathbf{x}_{t+1})$ is not, thereby rendering Equation (3) intractable. In practice, this is typically handled by replacing the conditional $q(\mathbf{x}_0|\mathbf{x}_{t+1})$ with a point estimate $\delta_{f_\theta(\mathbf{x}_{t+1})}(\mathbf{x}_0)$, where $f_\theta(\mathbf{x}_{t+1})$ is a "reconstruction" of $\mathbf{x}_0$, computed by a neural network. This is typically done with Tweedie's formula, where $f_\theta(\mathbf{x}_{t+1}) = \mathbf{x}_t + \sigma_{t+1}^2 s_\theta(\mathbf{x}_{t+1}, t+1) \approx \mathbb{E}[\mathbf{x}_0|\mathbf{x}_{t+1}]$ and $s_\theta(\mathbf{x}_{t+1}, t+1)$ is a neural network which approximates the score $\nabla_{\mathbf{x}_{t+1}} \log q(\mathbf{x}_{t+1})$ (and using the true score would result in the reconstruction being equal to the expected value). Plugging this approximation into Equation (3), the backward kernel becomes $p_\theta(\mathbf{x}_t|\mathbf{x}_{t+1}) := q(\mathbf{x}_t|\mathbf{x}_{t+1}, \mathbf{x}_0 = f_\theta(\mathbf{x}_{t+1}))$.

**Probability Flow ODE** Song et al. (2021b) described how diffusion models can be generalized as time-continuous stochastic differential equations (SDEs), and how sampling form a diffusion model can be viewed as solving the corresponding reverse-time SDE. They further derive a "probability flow ordinary differential equation (PF-ODE)", which allows sampling from the same distribution $p_\theta(\mathbf{x}_t)$ as the

SDE by solving a deterministic ODE (initialized randomly). The solution $\mathbf{x}_0$ is a sample from $p_\theta(\mathbf{x}_0)$, and this is used by Zhang et al. (2024) to construct an alternative transition kernel: first step is obtaining a sample $\hat{\mathbf{x}}_{0,t+1}$ by solving the ODE starting at $\mathbf{x}_{t+1}$, then sample from $q(\mathbf{x}_t|\mathbf{x}_0 = \hat{\mathbf{x}}_{0,t+1})$. This can give samples from the same marginals $p_\theta(\mathbf{x}_t)$ as if using the regular kernel $p_\theta(\mathbf{x}_t|\mathbf{x}_{t+1})$, but for unconditional sampling this is a convoluted way of performing generation as $\hat{\mathbf{x}}_{0,t+1}$ is already a sample from the desired distribution. However, as we discuss in Section 3, this approach does provide an interesting possibility for conditional sampling.

### 2.2. Sequential Monte Carlo

Sequential Monte Carlo (SMC; see, e.g., Naesseth et al., 2019, for an overview) is a class of methods that enable sampling from a sequence of distributions $\{\pi_t(\mathbf{x}_{t:T})\}_{t=0}^T$, which are known and can be evaluated up to a normalizing constant, i.e.,

$$\pi_t(\mathbf{x}_{t:T}) = \gamma_t(\mathbf{x}_{t:T})/Z_t, \tag{4}$$

where $\gamma_t(\mathbf{x}_{t:T})$ can be evaluated pointwise. In an SMC algorithm, a set of $N$ samples, or *particles*, are generated in parallel, and they are weighted such that a set of (weighted) particles $\{(\mathbf{x}_{t:T}^i, w_t^i)\}_{i=1}^N$ are approximate draws from the target distribution $\pi_t(\mathbf{x}_{t:T})$. For each $t$, an SMC algorithm consists of three steps. The *resampling* step samples a new set of particles $\{(\mathbf{x}_{t:T}^{a_t^i})\}_{i=1}^N$ with $a_t^1, \ldots, a_t^N \overset{\text{iid}}{\sim}$ Categorical($\{1, \ldots, N\}; \{w_t^i\}_{i=1}^N$)[3]. The second step is the *proposal* step, where new samples $\{\mathbf{x}_{t-1:T}^i\}_{i=1}^N$ are proposed as $\mathbf{x}_{t-1}^i \sim r_{t-1}(\mathbf{x}_{t-1}^i|\mathbf{x}_{t:T}^i)$, $\mathbf{x}_{t-1:T}^i = (\mathbf{x}_{t-1}^i, \mathbf{x}_{t:T}^i)$, and finally a *weighting* step,

$$w_{t-1}^i \propto \gamma_{t-1}(\mathbf{x}_{t-1:T}^i)/(r_{t-1}(\mathbf{x}_{t-1}^i|\mathbf{x}_{t:T}^i)\gamma_t(\mathbf{x}_{t:T}^i)). \tag{5}$$

To construct an SMC algorithm, it is hence necessary to determine two components: the target distributions $\{\pi_t(\mathbf{x}_{t:T})\}_{t=0}^T$ (or rather, the unnormalized distributions $\{\gamma_t(\mathbf{x}_{t:T})\}_{t=0}^T$), and the proposals $\{r_{t-1}(\mathbf{x}_{t-1}|\mathbf{x}_{t:T})\}_{t=1}^T$.

**SMC as a General Sampler** Even though SMC relies on a sequence of (unnormalized) distributions, it can still be used as a general sampler to sample from some "static" distribution $\phi(\mathbf{x})$ by introducing auxiliary variables $\mathbf{x}_{0:T}$ and *constructing* a sequence of distributions over $\{\mathbf{x}_{t:T}\}_{t=0}^T$. As long as the marginal distribution of $\mathbf{x}_0$ w.r.t. the *final* distribution $\pi_0(\mathbf{x}_{0:T})$ is equal to $\phi(\mathbf{x}_0)$, the SMC algorithm will provide a consistent approximation of $\phi(\mathbf{x})$ (i.e., increasingly accurate as the number of particles $N$ increases). The *intermediate target* distributions $\{\pi_t(\mathbf{x}_{t:T})\}_{t=1}^T$ are then merely a means for approximating the final target $\pi_0(\mathbf{x}_{0:T})$.

---

[2]In principle, $\mathbf{x}_t$ could also be multiplied by some time-dependent parameter, but we skip this here for simplicity.

[3]The resampling procedure is a design choice. In this paper we use Multinomial with probabilities $\{w_t^i\}_{i=1}^N$ for simplicity.

# 3. Decoupled Diffusion SMC

## 3.1. Target Distributions

The sequential nature of the generative diffusion model naturally suggests that SMC can be used for the Bayesian inverse problem, by constructing a sequence of target distributions based on Equation (2). This approach has been explored in several recent works (Wu et al., 2023; Cardoso et al., 2024, see also Section 4). However, Zhang et al. (2024) recently proposed an alternative simulation protocol for tackling inverse problems with diffusion priors, based on a "decoupling" argument: they propose to simulate (approximately) from $p_\theta(\mathbf{x}_0|\mathbf{x}_{t+1}, \mathbf{y})$ and then push this sample forward to diffusion time $t$ by sampling from the forward kernel $q(\mathbf{x}_t|\mathbf{x}_0)$ instead of $q(\mathbf{x}_t|\mathbf{x}_0, \mathbf{x}_{t+1})$. The motivation is to reduce the autocorrelation in the generative process to enable making transitions with larger updates and thus correct larger, global errors. The resulting method is referred to as Decoupled Annealing Posterior Sampling (DAPS).

To leverage this idea, we can realize that the SMC framework is in fact very general and, as discussed in Section 2.2, the sequence of target distributions can be seen as a design choice, as long as the final target $\pi_0(\mathbf{x}_{0:T})$ admits the distribution of interest as a marginal. Thus, it is possible to use the DAPS sampling protocol as a basis for SMC. This corresponds to redefining the prior over trajectories, from Equation (2) to $p_\theta^0(\mathbf{x}_{0:T}) = p_\theta(\mathbf{x}_T) \prod_{t=0}^{T-1} p_\theta^0(\mathbf{x}_t|\mathbf{x}_{t+1})$ where

$$p_\theta^0(\mathbf{x}_t|\mathbf{x}_{t+1}) = q(\mathbf{x}_t|\mathbf{x}_0 = f_\theta(\mathbf{x}_{t+1})). \quad (6)$$

Conceptually, this corresponds to reconstructing $\mathbf{x}_0$ conditionally on the current state $\mathbf{x}_{t+1}$, followed by adding noise to the reconstructed sample using the forward model. As discussed by Zhang et al. (2024), and also proven by us in Appendix A, the two transitions can still lead to the same marginal distributions for all time points $t$, i.e., $\int p_\theta^0(\mathbf{x}_{t:T})d\mathbf{x}_{t+1:T} = \int p_\theta(\mathbf{x}_{t:T})d\mathbf{x}_{t+1:T}$, under the assumption that $\mathbf{x}_0 = f_\theta(\mathbf{x}_{t+1})$ is a sample from $p_\theta(\mathbf{x}_0)$ and that the generative process $p_\theta$ indeed inverts the forward process $q$. The sample $\mathbf{x}_0$ can be approximately obtained by, e.g., solving the reverse-time PF-ODE, and using this solution as reconstruction model $f_\theta(\mathbf{x}_{t+1})$.

**Generalizing the DAPS Prior**  By rewriting the conditional forward kernel $q(\mathbf{x}_t|\mathbf{x}_{t+1}, \mathbf{x}_0)$ using Bayes theorem,

$$q(\mathbf{x}_t|\mathbf{x}_{t+1}, \mathbf{x}_0) \propto q(\mathbf{x}_{t+1}|\mathbf{x}_t)q(\mathbf{x}_t|\mathbf{x}_0), \quad (7)$$

we can view the standard diffusion backward kernel Equation (3) as applying the DAPS kernel, but conditioning the sample on the previous state $\mathbf{x}_{t+1}$, which acts as an "observation" with likelihood $q(\mathbf{x}_t|\mathbf{x}_{t+1})$. We can thus generalize the kernel in Equation (3) by annealing this likelihood with the inverse temperature $\eta$,

$$q_\eta(\mathbf{x}_t|\mathbf{x}_{t+1}, \mathbf{x}_0) \propto q(\mathbf{x}_{t+1}|\mathbf{x}_t)^\eta q(\mathbf{x}_t|\mathbf{x}_0), \quad (8)$$

and define our backward transition using this as

$$p_\theta^\eta(\mathbf{x}_t|\mathbf{x}_{t+1}) := \int q_\eta(\mathbf{x}_t|\mathbf{x}_{t+1}, \mathbf{x}_0)\delta_{f_\theta(\mathbf{x}_{t+1})}(\mathbf{x}_0)d\mathbf{x}_0 \quad (9)$$

which allows us to smoothly transition between the DAPS ($\eta = 0$) and standard ($\eta = 1$) backward kernels.

**Likelihood**  Having defined the prior as the generalized DAPS prior $p_\theta^\eta(\mathbf{x}_{t:T})$, we also need to incorporate the conditioning on the observation $\mathbf{y}$. A natural starting point would be to choose as targets

$$\gamma_t(\mathbf{x}_{t:T}) = p(\mathbf{y}|\mathbf{x}_t)p_\theta^\eta(\mathbf{x}_{t:T}), \quad (10)$$

as this for $t = 0$ leads to a distribution with marginal $p_\theta^\eta(\mathbf{x}|\mathbf{y})$. However, the likelihood $p(\mathbf{y}|\mathbf{x}_t) = \int p(\mathbf{y}|\mathbf{x}_0)p_\theta(\mathbf{x}_0|\mathbf{x}_t)d\mathbf{x}_0$ in Equation (10) is not tractable for $t > 0$, and needs to be approximated. As the reconstruction $f_\theta(\mathbf{x}_{t+1})$ played a central role in the prior, we can utilize this also for our likelihood. Song et al. (2023) proposed to use a Gaussian approximation $\tilde{p}_\theta(\mathbf{x}_0|\mathbf{x}_t) := \mathcal{N}(\mathbf{x}_0|f_\theta(\mathbf{x}_t), \rho_t^2 I)$, resulting in

$$p(\mathbf{y}|\mathbf{x}_t) \approx \tilde{p}(\mathbf{y}|\mathbf{x}_t) = \mathcal{N}(\mathbf{y}|Af_\theta(\mathbf{x}_t), \sigma_y^2 I + \rho_t^2 AA^T), \quad (11)$$

as the likelihood is linear and Gaussian. From hereon we will use the notation $\tilde{p}_\theta$ for any distribution derived from the Gaussian approximation $\tilde{p}_\theta(\mathbf{x}_0|\mathbf{x}_t)$. As for $t = 0$ the likelihood is known, we can rely on the consistency of SMC to obtain asymptotically exact samples, even if using approximate likelihoods in the intermediate targets.

**Putting it Together**  To summarize, we define a sequence of intermediate target distributions for SMC according to

$$\gamma_t(\mathbf{x}_{t:T}) = \tilde{p}(\mathbf{y}|\mathbf{x}_t)p_\theta(\mathbf{x}_T) \prod_{s=t}^{T-1} p_\theta^\eta(\mathbf{x}_s|\mathbf{x}_{s+1})$$

$$= \frac{\tilde{p}(\mathbf{y}|\mathbf{x}_t)}{\tilde{p}(\mathbf{y}|\mathbf{x}_{t+1})}p_\theta^\eta(\mathbf{x}_t|\mathbf{x}_{t+1})\gamma_{t+1}(\mathbf{x}_{t+1:T}), \quad (12)$$

where

$$\tilde{p}(\mathbf{y}|\mathbf{x}_t) = \mathcal{N}(\mathbf{y}|Af_\theta(\mathbf{x}_t), \sigma_y^2 I + \rho_t^2 AA^T) \quad (13)$$

and, for $\eta = 0$,[4]

$$p_\theta^{\eta=0}(\mathbf{x}_t|\mathbf{x}_{t+1}) = \mathcal{N}(\mathbf{x}_t|f_\theta(\mathbf{x}_{t+1}), \sigma_t^2 I). \quad (14)$$

We view the reconstruction $f_\theta$ as a design choice, and try Tweedie's formula and solving the PF-ODE initialized at $\mathbf{x}_{t+1}$ in the experiments section.

---

[4]See Appendix D for an expression for general $\eta$.

**Algorithm 1** Decoupled Diffusion SMC (DDSMC). All operations for $i = 1, \ldots, N$

---

**Input:** Score model $s_\theta$, measurement $\mathbf{y}$
**Output:** Sample $\mathbf{x}_0$
Sample $\mathbf{x}_T^i \sim p(\mathbf{x}_T)$
**for** $t$ in $T \ldots, 1$ **do**
  Predict $\hat{\mathbf{x}}_{0,t}^i = f_\theta(\mathbf{x}_t^i)$
  Compute $\tilde{p}(\mathbf{y}|\mathbf{x}_t^i)$ {*Equation* (13)}
  **if** $t = T$ **then**
    Set $\tilde{w}^i = \tilde{p}(\mathbf{y}|\mathbf{x}_T^i)$
  **else**
    Set $\tilde{w}^i = \frac{\tilde{p}(\mathbf{y}|\mathbf{x}_t^i)p_\theta^\eta(\mathbf{x}_t^i|\mathbf{x}_{t+1}^i)}{\tilde{p}(\mathbf{y}|\mathbf{x}_{t+1}^i)r_t(\mathbf{x}_t^i|\mathbf{x}_{t+1}^i,\mathbf{y})}$
  **end if**
  Compute $w^i = \tilde{w}^i / \sum_{j=1}^N \tilde{w}^j$
  Resample $\{\mathbf{x}_t^i, \hat{\mathbf{x}}_{0,t}^i\}_{i=1}^N$
  Sample $\mathbf{x}_{t-1}^i \sim r_{t-1}(\mathbf{x}_{t-1}|\mathbf{x}_t^i, \mathbf{y})$ {*Equation* (19)*, or Equation* (54) *for general* $\eta$.}
**end for**
Compute $\tilde{w}^i = \frac{p(\mathbf{y}|\mathbf{x}_0^i)}{p(\mathbf{y}|\mathbf{x}_1^i)}$ and $w^i = \tilde{w}^i / \sum_{j=1}^N \tilde{w}^j$
Sample $\mathbf{x}_0 \sim \text{Multinomial}(\{\mathbf{x}_0^j\}_{j=1}^N, \{w^j\}_{j=1}^N)$
**return** $\mathbf{x}_0$ {*Or the full set of weighted samples depending on application*}

---

**Proposition 3.1.** *The target in Equations* (12) *and* (13) *with* $\rho_0^2 = 0$ *will, under mild assumptions, give an SMC algorithm targeting* $p_\theta^\eta(\mathbf{x}_0|\mathbf{y}) \propto p(\mathbf{y}|\mathbf{x}_0)p_\theta^\eta(\mathbf{x}_0)$ *that is asymptotically exact, i.e., asymptotically unbiased when the number of particles increases.*

This follows from general SMC theory, as the unnormalized target at $t = 0$ becomes $\gamma_0(\mathbf{x}_{0:T}) = p(\mathbf{y}|\mathbf{x}_0)p_\theta(\mathbf{x}_{0:T})$, meaning the normalized target $\pi_0(\mathbf{x}_{0:T}) = \gamma_0(\mathbf{x}_{0:T})/Z_0$ will have $p_\theta(\mathbf{x}_0|\mathbf{y})$ as marginal, i.e., $\int \pi_0(\mathbf{x}_{0:T})d\mathbf{x}_{1:T} = p_\theta(\mathbf{x}_0|\mathbf{y})$. As SMC provides consistent approximations of all its target distributions $\{\pi_t(\mathbf{x}_{t:T})\}_{t=0}^T$, it will in particular give a consistent approximation at $t = 0$.

*Remark* 3.2. Whether the assumptions needed for the marginals of the DAPS ($\eta = 0$) and standard diffusion ($\eta = 1$) prior to coincide actually are fulfilled *does not affect the consistency in Proposition 3.1.* If they are not fulfilled, the priors will be different, but the consistency refers to consistent approximations of the posterior *induced by the chosen prior*. This is in contrast to FPS (Dou & Song, 2023) and SMCDiff (Trippe et al., 2023), which *rely on the assumption of the matching of the backward and forward processes for their target to coincide with the induced posterior*, and hence do not enjoy the asymptotical exactness guarantees unless this assumption is met.

## 3.2. Proposal

As the efficiency of the SMC algorithm in practice very much depends on the proposal, we will, as previous works on SMC (e.g., TDS (Wu et al., 2023) and MCGDiff (Cardoso et al., 2024)), use a proposal which incorporates information about the measurement $\mathbf{y}$. Again we are inspired by DAPS where the Gaussian approximation $\tilde{p}_\theta(\mathbf{x}_0|\mathbf{x}_t)$ plays a central role. In their method, they use this approximation as a prior over $\mathbf{x}_0$, which together with the likelihood $p(\mathbf{y}|\mathbf{x}_0)$ form an (approximate) posterior

$$\tilde{p}_\theta(\mathbf{x}_0|\mathbf{x}_{t+1}, \mathbf{y}) \propto p(\mathbf{y}|\mathbf{x}_0)\tilde{p}_\theta(\mathbf{x}_0|\mathbf{x}_{t+1}). \quad (15)$$

The DAPS method is designed for general likelihoods and makes use of Langevin dynamics to sample from this approximate posterior. However, in the linear-Gaussian case we can, similarly to Equation (11), obtain a closed form expression as

$$\tilde{p}_\theta(\mathbf{x}_0|\mathbf{x}_{t+1}, \mathbf{y}) = \mathcal{N}(\mathbf{x}_0|\tilde{\mu}_\theta^{t+1}(\mathbf{x}_{t+1}, \mathbf{y}), \mathbf{M}_{t+1}^{-1}), \quad (16)$$

with $\tilde{\mu}_\theta^{t+1}(\mathbf{x}_{t+1}, \mathbf{y}) = \mathbf{M}_{t+1}^{-1}\mathbf{b}_{t+1}$ and

$$\mathbf{M}_{t+1} = \frac{1}{\sigma_y^2}\mathbf{A}^T\mathbf{A} + \frac{1}{\rho_{t+1}^2}I, \quad (17a)$$

$$\mathbf{b}_{t+1} = \frac{1}{\sigma_y^2}\mathbf{A}^T\mathbf{y} + \frac{1}{\rho_{t+1}^2}f_\theta(\mathbf{x}_{t+1}). \quad (17b)$$

Although this is an approximation of the true posterior, it offers an interesting venue for an SMC proposal by propagating a sample from the posterior forward in time. Assuming $\eta = 0$ for notational brevity (see Appendix D for the general expression) we can write this as

$$r_t(\mathbf{x}_t|\mathbf{x}_{t+1}, \mathbf{y}) = \int \tilde{q}(\mathbf{x}_t|\mathbf{x}_0)\tilde{p}_\theta(\mathbf{x}_0|\mathbf{x}_{t+1}, \mathbf{y})d\mathbf{x}_0. \quad (18)$$

This proposal is similar to one step of the generative procedure used in DAPS, where in their case the transition would correspond to using the diffusion forward kernel, i.e., $\tilde{q}(\mathbf{x}_t|\mathbf{x}_0) = q(\mathbf{x}_t|\mathbf{x}_0)$. We make a slight adjustment based on the following intuition. Considering the setting of non-informative measurements, we expect our posterior $p_\theta(\mathbf{x}_0|\mathbf{y})$ to coincide with the prior. To achieve this, we could choose $\tilde{q}(\mathbf{x}_t|\mathbf{x}_0)$ such that $r_t(\mathbf{x}_t|\mathbf{x}_{t+1}) = p_\theta^0(\mathbf{x}_t|\mathbf{x}_{t+1})$. Using $q(\mathbf{x}_t|\mathbf{x}_0)$ together with the prior $\tilde{p}_\theta(\mathbf{x}_0|\mathbf{x}_{t+1})$ will, however, not match the covariance in $p_\theta^0(\mathbf{x}_t|\mathbf{x}_{t+1})$, since the Gaussian approximation $\tilde{p}_\theta(\mathbf{x}_0|\mathbf{x}_{t+1})$ will inflate the variance by a term $\rho_{t+1}^2$. To counter this effect, we instead opt for a covariance $\lambda_t^2 I$ (i.e., $\tilde{q}(\mathbf{x}_t|\mathbf{x}_0) = \mathcal{N}(\mathbf{x}_t|\mathbf{x}_0, \lambda_t^2 I)$) such that in the unconditional case, $r_t(\mathbf{x}_t|\mathbf{x}_{t+1}) = p_\theta^0(\mathbf{x}_t|\mathbf{x}_{t+1})$. As everything is Gaussian, the marginalization in Equation (18) can be computed exactly, and we obtain our proposal for $t > 0$ as

$$r_t(\mathbf{x}_t|\mathbf{x}_{t+1}, \mathbf{y}) = \mathcal{N}(\mathbf{x}_t|\tilde{\mu}_\theta^{t+1}(\mathbf{x}_{t+1}, \mathbf{y}), \lambda_t^2 I + \mathbf{M}_{t+1}^{-1}) \quad (19)$$

where $\lambda_t^2 = \sigma_t^2 - \rho_{t+1}^2$ and, for $t = 0$, $r_0(\mathbf{x}_0|\mathbf{x}_1, \mathbf{y}) = \delta_{\tilde{\mu}_\theta^1(\mathbf{x}_1, \mathbf{y})}(\mathbf{x}_0)$. We can directly see that in the non-informative case (i.e., $\sigma_y \to \infty$), we will recover the prior as $M_{t+1}^{-1} = \rho_{t+1}^2 I$ and $\mathbf{b}_{t+1} = \frac{1}{\rho_{t+1}^2} f_\theta(\mathbf{x}_{t+1})$.

It can be noted that, although the proposal relies on approximations (like the approximate posterior in Equation (16)), the SMC framework still provides an asymptotically exact algorithm by targeting the desired posterior $p_\theta(\mathbf{x}_{0:T}|\mathbf{y})$, as established in Proposition 3.1. Approximations in the proposal do not affect that property.

If using the generalized DAPS prior from the previous section, we can also replace $\tilde{q}(\mathbf{x}_t|\mathbf{x}_0)$ with $\tilde{q}_\eta(\mathbf{x}_t|\mathbf{x}_{t+1}, \mathbf{x}_0)$ (cf. Equation (8)) and make the corresponding matching of covariance.

**Detailed Expressions for Diagonal $A$**   If first assuming that $A$ is diagonal in the sense that it is of shape $d_y \times d_x$ ($d_y \leq d_x$) with non-zero elements $(a_1, \ldots, a_{d_y})$ only along the main diagonal, we can write out explicit expressions for the proposal. In this case, the covariance matrix $M_{t+1}^{-1}$ is diagonal with the $i$:th diagonal element ($i = 1, \ldots, d_x$) becoming

$$(M_{t+1}^{-1})_{ii} = \begin{cases} \frac{\rho_{t+1}^2 \sigma_y^2}{a_i^2 \rho_{t+1}^2 + \sigma_y^2} & i \leq d_y \\ \rho_{t+1}^2 & i > d_y \end{cases}. \quad (20)$$

This means that the covariance matrix in the proposal is diagonal with the elements $\sigma_t^2 - \rho_{t+1}^2 + \frac{\rho_{t+1}^2 \sigma_y^2}{a_i^2 \rho_{t+1}^2 + \sigma_y^2}$ for $i \leq d_y$, and $\sigma_t^2$ for $i > d_y$. Further, the mean of the $i$:th variable in the proposal becomes

$$\tilde{\mu}_\theta^{t+1}(\mathbf{x}_{t+1}, \mathbf{y})_i =$$
$$\begin{cases} \frac{a_i \rho_{t+1}^2}{a_i^2 \rho_{t+1}^2 + \sigma_y^2} y_i + \frac{\sigma_y^2}{a_i^2 \rho_{t+1}^2 + \sigma_y^2} f_\theta(\mathbf{x}_{t+1})_i & i \leq d_y \\ f_\theta(\mathbf{x}_{t+1})_i & i > d_y. \end{cases} \quad (21)$$

We can see the effect of the decoupling clearly by considering the noise-less case, $\sigma_y = 0$. In that case, we are always using the known observed value $(\mathbf{x}_0)_i = y_i/a_i$ as the mean, not taking the reconstruction, nor the previous value $(\mathbf{x}_{t+1})_i$, into account.

As the proposal is a multivariate Gaussian with diagonal covariance matrix, we can efficiently sample from it using independent samples from a standard Gaussian. We provide expressions for general choices of $\eta$ in Appendix D.

**Non-diagonal $A$**   For a general non-diagonal $A$, the proposal at first glance looks daunting as it requires inverting $M_{t+1}$ which is a (potentially very large) non-diagonal matrix, and sampling from a multivariate Gaussian with non-diagonal covariance matrix (which requires a computationally expensive Cholesky decomposition). However,

similar to previous work (Kawar et al., 2021; 2022; Cardoso et al., 2024), by writing $A$ in terms of its singular value decomposition $A = USV^T$ where $U \in \mathbb{R}^{d_y \times d_y}$ and $V \in \mathbb{R}^{d_x \times d_x}$ are orthonormal matrices, and $S$ a $d_y \times d_x$-dimensional matrix with non-zero elements only on its main diagonal, we can multiply both sides of the measurement equation by $U^T$ to obtain

$$U^T \mathbf{y} = SV^T \mathbf{x} + \sigma_y U^T \epsilon. \quad (22)$$

By then defining $\mathbf{y}' = U^T \mathbf{y}$, $\mathbf{x}' = V^T \mathbf{x}$, $A' = S$, and $\epsilon' = U^T \epsilon \sim \mathcal{N}(0, UIU^T) = \mathcal{N}(0, I)$, we obtain the new measurement equation

$$\mathbf{y}' = A'\mathbf{x}' + \sigma_y \epsilon', \quad \epsilon' \sim \mathcal{N}(0, I). \quad (23)$$

We can now run our DDSMC algorithm in this new basis and use the expressions from the diagonal case by replacing all variables with their corresponding primed versions, enabling efficient sampling also for non-diagonal $A$, given an implementable SVD. An algorithm outlining an implementation of DDSMC can be found in Algorithm 1.

### 3.3. D3SMC – A Discrete Version

We also extend our DDSMC algorithm to a discrete setting which we call Discrete Decoupled Diffusion SMC (D3SMC). Using $\mathbf{x}$ to denote a one-hot encoding of a single variable, we target the case where the measurement can be described by a transition matrix $Q_y$ as

$$p(\mathbf{y}|\mathbf{x}_0) = \text{Categorical}(\mathbf{p} = \mathbf{x}_0 Q_y), \quad (24)$$

and when the data consists of $D$ variables, the measurement factorizes over these variables (i.e., for each of the $D$ variables, we have a corresponding measurement). This setting includes both inpainting (a variable is observed with some probability, otherwise it is in an "unknown" state) and denoising (a variable randomly transitions into a different state). To tackle this problem, we explore the D3PM-uniform model (Austin et al., 2021) and derive a discrete analogy to DDSMC which uses the particular structure of D3PM. We elaborate on the D3PM model in general and our D3SMC algorithm in particular in Appendix E.

## 4. Related Work

**SMC and Diffusion Models**   The closest related SMC methods for Bayesian inverse problems with diffusion priors are the Twisted Diffusion Sampler (TDS) (Wu et al., 2023) and Monte Carlo Guided Diffusion (MCGDiff) (Cardoso et al., 2024). TDS is a general approach for solving inverse problems, while MCGDiff specifically focuses on the linear-Gaussian setting. These methods differ in the choices of intermediate targets and proposals. TDS makes use of the reconstruction network to approximate the likelihood at time $t$ as $p(\mathbf{y} \mid \mathbf{x}_0 = f_\theta(\mathbf{x}_t))$ and then add the

*Table 1.* Results on the Gaussian mixture model experiment when using DDSMC and reconstructing $\mathbf{x}_0$ using either Tweedie's formula ($f_\theta(\mathbf{x}_{t+1}) = \mathbb{E}[\mathbf{x}_0|\mathbf{x}_t]$) or solving the PF-ODE. Metric is the sliced Wasserstein distance between true posterior samples and samples from DDSMC, averaged over 20 seeds with 95% CLT confidence intervals.

| | | TWEEDIE | | | PF-ODE | | |
|---|---|---|---|---|---|---|---|
| $d_x$ | $d_y$ | $\eta = 0.0$ | $\eta = 0.5$ | $\eta = 1.0$ | $\eta = 0.0$ | $\eta = 0.5$ | $\eta = 1.0$ |
| | 1 | $1.90 \pm 0.48$ | $1.78 \pm 0.44$ | $1.57 \pm 0.43$ | $1.15 \pm 0.24$ | $0.88 \pm 0.33$ | $1.01 \pm 0.39$ |
| 8 | 2 | $0.69 \pm 0.27$ | $0.64 \pm 0.29$ | $0.50 \pm 0.27$ | $0.56 \pm 0.23$ | $0.34 \pm 0.20$ | $0.39 \pm 0.22$ |
| | 4 | $0.28 \pm 0.04$ | $0.22 \pm 0.03$ | $0.13 \pm 0.05$ | $0.21 \pm 0.10$ | $0.09 \pm 0.08$ | $0.17 \pm 0.06$ |
| | 1 | $1.26 \pm 0.34$ | $1.07 \pm 0.31$ | $0.96 \pm 0.28$ | $0.69 \pm 0.17$ | $0.50 \pm 0.16$ | $0.77 \pm 0.12$ |
| 80 | 2 | $1.14 \pm 0.40$ | $0.92 \pm 0.36$ | $0.81 \pm 0.35$ | $0.69 \pm 0.36$ | $0.37 \pm 0.20$ | $0.69 \pm 0.15$ |
| | 4 | $0.66 \pm 0.29$ | $0.50 \pm 0.27$ | $0.44 \pm 0.22$ | $0.41 \pm 0.23$ | $0.21 \pm 0.13$ | $0.54 \pm 0.04$ |
| | 1 | $1.73 \pm 0.54$ | $1.64 \pm 0.61$ | $1.86 \pm 0.63$ | $1.37 \pm 0.43$ | $1.68 \pm 0.52$ | $2.09 \pm 0.51$ |
| 800 | 2 | $1.06 \pm 0.56$ | $0.90 \pm 0.65$ | $1.34 \pm 0.68$ | $0.81 \pm 0.40$ | $1.17 \pm 0.65$ | $1.76 \pm 0.67$ |
| | 4 | $0.35 \pm 0.08$ | $0.16 \pm 0.10$ | $0.56 \pm 0.19$ | $0.23 \pm 0.06$ | $0.47 \pm 0.22$ | $1.24 \pm 0.47$ |

*Figure 1.* Samples from DDSMC from the GMM experiments ($d_x = 800$ and $d_y = 1$) using either Tweedie's formula (three left-most figures) or the solution of the PF-ODE (three right-most figures) as a reconstruction, and using different values of $\eta$ in the generalized DAPS prior. Blue samples are from the posterior, while red samples are from DDSMC. More examples can be found in Appendix G.1.

score of this approximate likelihood as a drift in the transition kernel. This requires differentiating through the reconstruction model w.r.t. $\mathbf{x}_t$, which can incur a significant computational overhead. MCGDiff instead uses the forward diffusion model to push the observation $\mathbf{y}$ "forward in time". Specifically, they introduce a potential function at time $t$ which can be seen as a likelihood corresponding to the observation model $\hat{\mathbf{y}}_t = A\mathbf{x}_t$, where $\hat{\mathbf{y}}_t$ is a noised version (according to the forward model at time $t$) of the original observation $\mathbf{y}$. DDSMC differs from both of these methods, and is conceptually based on reconstructing $\mathbf{x}_0$ from the current state $\mathbf{x}_t$, performing an explicit conditioning on $\mathbf{y}$ at time $t = 0$, and then pushing the *posterior distribution* forward to time $t - 1$ using the forward model. We further elaborate on the differences between TDS, MCGDiff, and DDSMC in Appendix C (see also the discussion by Zhao et al. (2024)).

SMCDiff (Trippe et al., 2023) and FPS (Dou & Song, 2023) are two other SMC algorithms that target posterior sampling with diffusion priors, but these rely on the assumption that the learned backward process is an exact reversal of the forward process, and are therefore not consistent in general. SMC has also been used as a type of *discriminator* guidance (Kim et al., 2023) of diffusion models in both the continuous (Liu et al., 2024) and discrete (Ekström Kelvinius & Lindsten, 2024) setting.

**Posterior Sampling with Diffusion Priors** The closest non-SMC work to ours is of course DAPS (Zhang et al., 2024), which we build upon, but also generalize in several ways. DDSMC is also related to ΠGDM (Song et al., 2023), which introduces the Gaussian approximation used in Equations (11) and (15). Other proposed methods include DDRM (Kawar et al., 2022) which defines a task-specific *conditional* diffusion process which depends on a reconstruction $f_\theta(\mathbf{x}_t)$, but where the optimal solution can be approximated with a model trained on the regular unconditional task. Recently, Janati et al. (2024) proposed a method referred to as DCPS, which also builds on the notion of intermediate targets, but not within an SMC framework. Instead, to sample from these intermediate targets, they make use of Langevin sampling and a variational approximation that is optimized with stochastic gradient descent within each step of the generative model. All of these methods include various approximations, and contrary to DDSMC, none of them provides a consistent approximation of a given posterior. Other sampling methods include MCMC methods like Gibbs sampling (Coeurdoux et al., 2024; Wu et al., 2024).

*Table 2.* Comparison of DDSMC with other methods in the Gaussian mixture setting. Numbers for DDSMC are the best numbers from Table 1. *All methods have been run for 20 different seeds, but TDS shows instability and crashes, meaning that their numbers are computed over fewer runs.

| $d_x$ | $d_y$ | DDSMC | MCGDIFF | TDS* | DCPS | DDRM | DAPS |
|---|---|---|---|---|---|---|---|
| 8 | 1 | $0.88 \pm 0.33$ | $1.79 \pm 0.54$ | $9.72 \pm 9.89$ | $2.51 \pm 1.02$ | $3.83 \pm 1.05$ | $5.63 \pm 0.90$ |
|  | 2 | $0.34 \pm 0.20$ | $0.80 \pm 0.41$ | $5.23 \pm 2.57$ | $1.35 \pm 0.70$ | $2.25 \pm 0.92$ | $5.93 \pm 1.16$ |
|  | 4 | $0.09 \pm 0.08$ | $0.26 \pm 0.21$ | $3.02 \pm 2.58$ | $0.44 \pm 0.25$ | $0.55 \pm 0.29$ | $4.85 \pm 1.34$ |
| 80 | 1 | $0.50 \pm 0.16$ | $1.06 \pm 0.36$ | $7.37 \pm 7.62$ | $1.19 \pm 0.41$ | $5.19 \pm 1.07$ | $6.85 \pm 1.16$ |
|  | 2 | $0.37 \pm 0.20$ | $1.04 \pm 0.49$ | $2.63 \pm 1.40$ | $1.10 \pm 0.55$ | $5.62 \pm 1.09$ | $8.49 \pm 0.92$ |
|  | 4 | $0.21 \pm 0.13$ | $0.80 \pm 0.42$ | $1.47 \pm 1.42$ | $0.56 \pm 0.24$ | $4.95 \pm 1.25$ | $9.04 \pm 0.74$ |
| 800 | 1 | $1.37 \pm 0.43$ | $1.64 \pm 0.48$ | $2.45 \pm 0.96$ | $2.45 \pm 0.58$ | $7.15 \pm 1.11$ | $7.03 \pm 1.20$ |
|  | 2 | $0.81 \pm 0.40$ | $1.29 \pm 0.66$ | $2.84 \pm 1.32$ | $2.80 \pm 0.91$ | $8.21 \pm 0.88$ | $8.31 \pm 1.01$ |
|  | 4 | $0.16 \pm 0.10$ | $1.21 \pm 0.94$ | $3.58 \pm 3.18$ | $1.84 \pm 0.76$ | $8.66 \pm 0.87$ | $9.21 \pm 0.83$ |

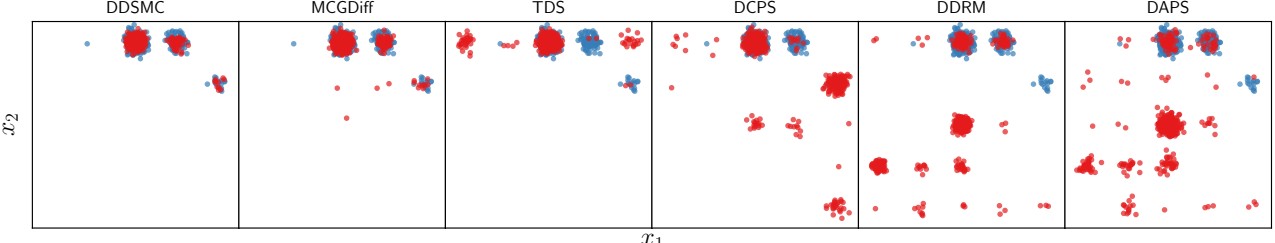

*Figure 2.* Qualitative comparison between DDSMC (using PF-ODE as reconstruction and $\eta = 0$) and other methods on the GMM experiments, $d_x = 800$ and $d_y = 1$. More qualitative comparisons can be found in Appendix G.1

## 5. Experiments

### 5.1. Gaussian Mixture Model

We first experiment on synthetic data, and use the Gaussian mixture model problem from Cardoso et al. (2024). Here, the prior on $\mathbf{x}$ is a Gaussian mixture, and both the posterior and score are therefore known on closed form[5] (we give more details in Appendix G.1), enabling us to verify the efficiency of DDSMC while ablating all other errors. As a metric, we use the sliced Wasserstein distance (Flamary et al., 2021) between 10k samples from the true posterior and each sampling algorithm.

We start with investigating the influence of $\eta$ and the reconstruction function $f_\theta$. We run DDSMC with $N = 256$ particles, and use $T = 20$ steps in the generative process. As a reconstruction, we compare Tweedie's formula and the DDIM ODE solver (Song et al., 2021a), see Equation (78) in the appendix, where we solve the ODE using as many steps as there are "left" in the diffusion process. In Table 1, we see that using Tweedie's reconstruction requires a larger value of $\eta$. This can be explained by the fact that $\eta = 0$ (DAPS prior) requires exact samples from $p_\theta(\mathbf{x}_0|\mathbf{x}_t)$ for the DAPS

---

[5]This includes using the DDPM (Ho et al., 2020) or "Variance-preserving" formulation of a diffusion model. We have included a derivation of DDSMC for this setting in Appendix B

*Table 3.* LPIPS results on FFHQ experiments. The noise level is $\sigma_y = 0.05$, and the tasks are inpainting a box in the **mid-dle**, outpainting right **half** of the image, and **s**uper-**r**esolution $4\times$. Numbers are averages over 1k images. **Lower is better.**

|  |  | MIDDLE | HALF | SR4 |
|---|---|---|---|---|
|  | DDRM | 0.04 | 0.25 | 0.20 |
|  | DCPS | 0.03 | 0.20 | 0.10 |
|  | DAPS | 0.05 | 0.24 | 0.15 |
|  | MCGDIFF | 0.10 | 0.34 | 0.15 |
| TWEEDIE | DDSMC-0.0 | 0.07 | 0.26 | 0.27 |
|  | DDSMC-0.5 | 0.07 | 0.27 | 0.20 |
|  | DDSMC-1.0 | 0.05 | 0.24 | 0.14 |
| ODE | DDSMC-0.0 | 0.05 | 0.23 | 0.21 |
|  | DDSMC-0.5 | 0.05 | 0.23 | 0.15 |
|  | DDSMC-1.0 | 0.08 | 0.4 | 0.36 |

prior to agree with the original prior (see Appendix A), and this distribution is more closely approximated using the ODE reconstruction. For higher dimensions (i.e., $d_x = 800$) and using $\eta > 0$, Tweedie's formula works slightly better than the ODE solver. Qualitatively, we find that using a smaller $\eta$ and/or using Tweedie's reconstruction tends to concentrate the samples around the different modes, see Figure 1 for an example of $d_x = 800$ and $d_y = 1$ (additional examples in Appendix G.1). This plot also shows how, in

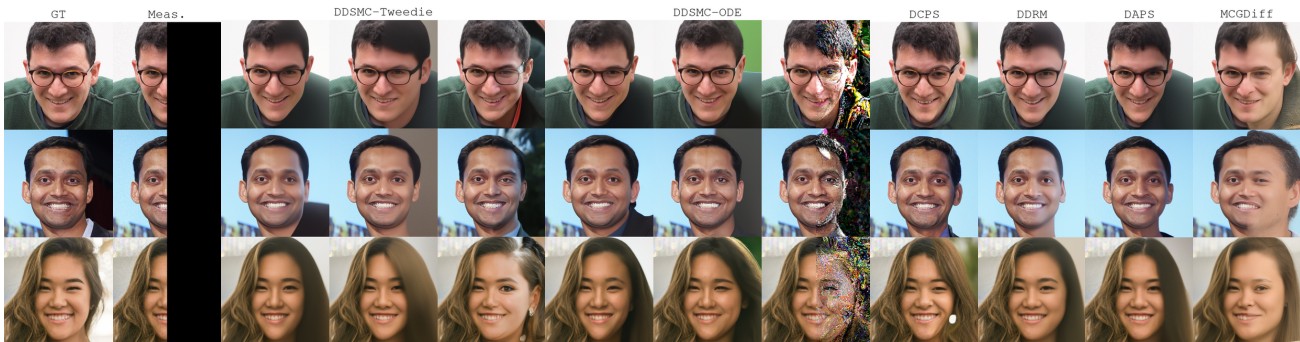

*Figure 3.* Examples of generated images for the outpainting task. The DDSMC samples are ordered with $\eta = 0$ to the left, $\eta = 0.5$ in the middle, and $\eta = 1$ to the right. Ground truth images are FFHQ images ID 100, 103, and 110 (see Table 11 in the appendix for attribution).

high dimensions, using a smaller $\eta$ is preferable, which could be attributed to lower $\eta$ enabling larger updates necessary in higher dimensions. It hence seems to be an interplay between the data dimensionality, the inverse temperature $\eta$, and the reconstruction function.

Next, we compare DDSMC with other methods, both SMC-based (MCGDiff and TDS) and non-SMC-based (DDRM, DCPS, DAPS). For all SMC methods, we use 256 particles, and accelerated sampling according to DDIM (Song et al., 2021a) with 20 steps, which we also used for DAPS to enabling evaluating the benefits of our generalization of the DAPS method without confounding the results with the effect of common hyperparameters. For DDRM and DCPS, however, we used 1 000 steps. More details are available in Appendix G.1. We see quantitatively in Table 2 that DDSMC outperforms all other methods, even when using Tweedie's reconstruction (which requires the same amount of compute as MCGDiff). We provide additional results with fewer particles in the appendix. As our *proposal* is essentially equal to DAPS (if using $\lambda^2 = \sigma^2$ in Equation (19)), running DDSMC with *a single particle* is essentially equal to DAPS (we verify this empirically in Appendix G.1.4). Therefore, we can conclude that it is indeed the introduction of SMC (multiple particles and resampling) that contributes to the improved performance over DAPS. A qualitative inspection of generated samples in Figure 2 shows how DDSMC is the most resistant method to sampling from spurious modes, while DAPS struggles to sample from the posterior. These are general trends we see when repeating with different seeds, see Appendix G.1.

### 5.2. Image Restoration

We now turn our attention to the problem of image restoration, and use our method for inpainting, outpainting, and super-resolution on the FFHQ (Karras et al., 2019) dataset (downsampled to $256 \times 256$), using a pretrained model by Chung et al. (2023). We implement DDSMC in the code-

base of DCPS[6], and use 1k images from the validation set and compute LPIPS (Zhang et al., 2018) as a quantitative metric (see PSNR results in Appendix G.2). The results can be found in Table 3, in addition to numbers for DDRM, DCPS, DAPS, and MCGDiff. Standard deviations can be found in Appendix G.2. We used 5 particles for our method, and when using the PF-ODE as reconstruction, we used 5 ODE steps. Further implementation details are available in Appendix G.2. It should be noted that LPIPS measures perceptual similarity between the sampled image and the ground truth, which is not the same as a measurement of how well the method samples from the true posterior. Specifically, it does not say anything about diversity of samples or how well the model captures the posterior uncertainty, which we could verify in the GMM experiment. Nevertheless, the numbers indicate that image reconstruction is on par with previous methods. Notably, MCGDiff, which is also an SMC method and have the same asymptotic guarantees in terms of posterior sampling, performs similar or worse to DDSMC.

Inspecting the generated images, we can see a visible effect when altering the reconstruction function and $\eta$, see Figure 3 and more examples in Appendix G.2. It seems like increasing $\eta$ and/or using the PF-ODE as a reconstruction function adds more details to the image. With the GMM experiments in mind, where we saw that lower $\eta$ and using Tweedie's formula as reconstruction function made samples more concentrated around the modes, it can be argued that there is a similar effect here: using Tweedie's formula and lower $\eta$ lead to sampled images closer to a "mode" of images, meaning details are averaged out. On the other hand, changing to ODE-reconstruction and increasing $\eta$ further away from the mode, which corresponds to more details in the images. For the case of ODE and $\eta = 1.0$, we see clear artifacts which explains the poor quantitative results.

---

[6]https://github.com/Badr-MOUFAD/dcps/

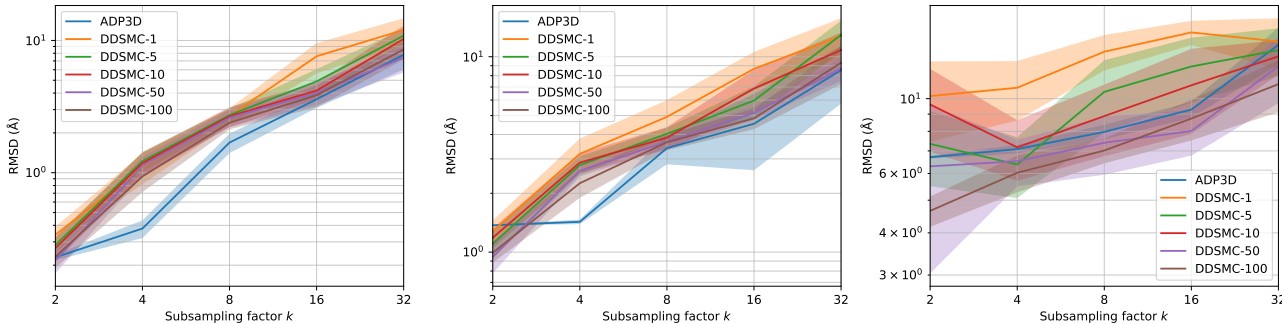

Figure 4. RMSD vs subsampling factor $k$ for the protein structure completion problem on the ATAD2 protein (Davison et al., 2022) (PDB identifier `7qum` (Turberville et al., 2023)) and different noise levels $\sigma$: 0 (left), 0.1 (middle), 0.5 (right). Comparing ADP-3D (Levy et al., 2024) and DDSMC with different number of particles.

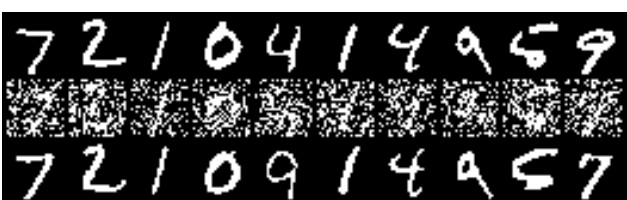

Figure 5. Qualitative results on binary MNIST using the discrete counterpart of DDSMC, D3SMC. Top is the ground truth, middle is measurement, and bottom is sampled image. More results can be found in Appendix G.4

### 5.3. Protein Structure Completion

As a different type of problem, we test DDMSC on the protein structure completion example from Levy et al. (2024), where we try to predict the coordinates of all heavy atoms in the backbone, observing coordinates of all 4 heavy atoms in the backbone of every $k$ residue. More details are in Appendix F. Data used is from the Protein Data Bank[7] (Berman et al., 2000; 2003). Figure 4 shows an example of the RMSD for different noise levels $\sigma$ and subsampling factors $k$. Especially on high noise levels, we are close to or better than ADP-3D (Levy et al., 2024), and we also see clear effects of introducing more particles, indicating the effectiveness of introducing the SMC aspect. Experiment details and more results are in Appendix G.3.

### 5.4. Discrete Data

As a proof of concept of our discrete algorithm D3SMC (see detailed description in Appendix E), we try denoising on binarized MNIST (Deng, 2012) (i.e., each pixel is either 0 or 1), cropped to $24 \times 24$ pixels to remove padding. As a backbone neural network, we used a U-Net (Ronneberger et al., 2015) trained with cross-entropy loss. We use a measurement model $\mathbf{y} = \mathbf{x}Q_y$ with $Q_y = (1 - \beta_y)I + \beta_y \mathbb{1}\mathbb{1}^T/d$

and $\beta_y = 0.6$, i.e., the observed pixel has the same value as the original image with probability $0.7$, and opposite with probability $0.3$. We present qualitative results for D3SMC with $N = 5$ particles in Figure 5, and more qualitative results can be found in Appendix G.4. While the digits are often recovered, there are cases when they are not (e.g., a 4 becoming a 9). Looking at multiple draws in the appendix, it seems the model in such cases can sample different digits, suggesting a multi-modal nature of the posterior.

## 6. Discussion & Conclusion

We have designed an SMC algorithm, DDSMC, for posterior sampling with diffusion priors which we also extended to discrete data. The method is based on decoupled diffusion, which we generalize by introducing a hyperparameter that allows bridging between full decoupling and standard diffusion (no decoupling). We demonstrate the superior performance of DDSMC compared to the state-of-the-art on a synthetic problem, which enables a quantitative evaluation of how well the different methods approximate the true posterior. We additionally test DDSMC on image reconstruction and protein structure completion where it performs on par with previous methods. In the image case, it performs slightly better than the alternative SMC method MCGDiff, while underperforming compared to DCPS. However, while LPIPS indeed is a useful metric for evaluating image reconstruction, it does not inherently capture posterior approximation quality, which is the aim of our method. We found that the methods performing particularly well on image reconstruction struggle on the synthetic task, highlighting a gap between perceived image quality and ability to approximate the true posterior. As DDSMC still performs on par with previous work on images and proteins, and at the same time showing excellent performance on the GMM task, we view DDSMC as a promising method for bridging between solving challenging high-dimensional inverse problems and maintaining exact posterior sampling capabilities.

---

[7]`www.rcsb.org/`, `www.wwpdb.org/`

## Acknowledgments

This research was supported by Swedish Research Council (VR) grant no. 2020-04122, 2024-05011, the Wallenberg AI, Autonomous Systems and Software Program (WASP) funded by the Knut and Alice Wallenberg Foundation, and the Excellence Center at Linköping–Lund in Information Technology (ELLIIT) . Computations were enabled by the Berzelius resource provided by the Knut and Alice Wallenberg Foundation at the National Supercomputer Centre.

## Impact Statement

This paper presents work whose goal is to advance the field of Machine Learning. There are many potential societal consequences of our work, none which we feel must be specifically highlighted here.

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

# A. Proofs

## A.1. Equality between marginals

The marginal distributions $p_\theta^0(\mathbf{x}_t)$ and $p_\theta(\mathbf{x}_t)$ are equal, as follows from the following proposition, inspired by Zhang et al. (2024).

**Proposition A.1.** *Conditioning on a sample $\mathbf{x}_t \sim p(\mathbf{x}_t)$ to sample $\mathbf{x}_0 \sim p(\mathbf{x}_0|\mathbf{x}_t)$, and then forgetting about $\mathbf{x}_t$ when sampling $\mathbf{x}_{t-1} \sim p(\mathbf{x}_{t-1}|\mathbf{x}_0)$, leads to a draw from $p(\mathbf{x}_{t-1})$.*

*Proof.*

$$p(\mathbf{x}_{t-1}) = \int \underbrace{p(\mathbf{x}_{t-1}|\mathbf{x}_0)}_{(*)} \underbrace{p(\mathbf{x}_0)}_{(**)} d\mathbf{x}_0 = \tag{25}$$

$$= \int \underbrace{p(\mathbf{x}_{t-1}|\mathbf{x}_0)}_{(*)} \underbrace{\int p(\mathbf{x}_0|\mathbf{x}_t)p(\mathbf{x}_t)d\mathbf{x}_t}_{(**)} d\mathbf{x}_0 \tag{26}$$

$$= \int \int p(\mathbf{x}_{t-1}|\mathbf{x}_0)p(\mathbf{x}_0|\mathbf{x}_t)p(\mathbf{x}_t)d\mathbf{x}_0 d\mathbf{x}_t \tag{27}$$

$$= \mathbb{E}_{\mathbf{x}_t}\mathbb{E}_{\mathbf{x}_0|\mathbf{x}_t}p(\mathbf{x}_{t-1}|\mathbf{x}_0). \tag{28}$$

$\square$

*Remark* A.2. In practice, the forward process is given by $q$, meaning $p(\mathbf{x}_{t-1}|\mathbf{x}_0)$ would be replaced by $q(\mathbf{x}_{t-1}|\mathbf{x}_0)$ above. Hence, for the above proposition to hold, it assumes that the backward process is the inverse of the forward process (i.e., $p(\mathbf{x}_{t-1}|\mathbf{x}_0) = q(\mathbf{x}_{t-1}|\mathbf{x}_0)$).

*Remark* A.3. Proposition A.1 is the same as the proposition in (Zhang et al., 2024), but without conditioning on $\mathbf{y}$. Our proof can be used also for the conditional case (by only adding the corresponding conditioning on $\mathbf{y}$), but is different from the one by Zhang et al. (2024) as it does not rely on the assumption of their graphical model.

# B. DDSMC for DDPM/VP diffusion models

In the DDPM (Ho et al., 2020) or Variance-preserving (VP) (Song et al., 2021b) formulation of a diffusion model, the forward process can described by

$$q(\mathbf{x}_{t+1}|\mathbf{x}_t) = \mathcal{N}(\mathbf{x}_{t+1}|\sqrt{1 - \beta_{t+1}}\mathbf{x}_t, \beta_{t+1}I), \tag{29}$$

which leads to

$$q(\mathbf{x}_t|\mathbf{x}_0) = \mathcal{N}(\mathbf{x}_t|\sqrt{\bar{\alpha}_t}\mathbf{x}_0, (1 - \bar{\alpha}_t)I), \tag{30}$$

where $\bar{\alpha}_t = \prod_{s=1}^t \alpha_s$ with $\alpha_s = 1 - \beta_s$.

## B.1. Target distribution

The target distribution changes as the transition $p_\theta^0(\mathbf{x}_t|\mathbf{x}_{t+1}) = \int q(\mathbf{x}_t|\mathbf{x}_0)\delta_{f_\theta(\mathbf{x}_{t+1})}(\mathbf{x}_0)d\mathbf{x}_0$ changes when $q(\mathbf{x}_t|\mathbf{x}_0)$ changes. The new expression is now $p_\theta^0(\mathbf{x}_t|\mathbf{x}_{t+1}) = \mathcal{N}(\sqrt{\bar{\alpha}_t}f_\theta(\mathbf{x}_{t+1}), (1 - \bar{\alpha}_t)I)$

## B.2. Proposal

In the DDPM formulation, we reuse the same Gaussian prior over $\mathbf{x}_0$ given $\mathbf{x}_t$ as before, $\tilde{p}_\theta(\mathbf{x}_0|\mathbf{x}_t) = \mathcal{N}(f_\theta(\mathbf{x}_t), \rho_t^2 I)$ and hence the same posterior $\tilde{p}_\theta(\mathbf{x}_0|\mathbf{x}_{t+1}, \mathbf{y}) = \mathcal{N}(\tilde{\mu}_\theta^{t+1}(\mathbf{x}_{t+1}, \mathbf{y}), M_{t+1}^{-1})$. However, we use $\tilde{q}(\mathbf{x}_t|\mathbf{x}_0) = \mathcal{N}(\mathbf{x}_t|\sqrt{\bar{\alpha}_t}\mathbf{x}_t, \lambda_t^2 I)$ which means that the proposal becomes

$$r_t(\mathbf{x}_t|\mathbf{x}_{t+1}, \mathbf{y}) = \int \tilde{q}(\mathbf{x}_t|\mathbf{x}_0)\tilde{p}_\theta(\mathbf{x}_0|\mathbf{x}_{t+1}, \mathbf{y})d\mathbf{x}_0 \tag{31}$$

$$= \mathcal{N}(\sqrt{\bar{\alpha}_t}\tilde{\mu}_\theta^{t+1}(\mathbf{x}_{t+1}, \mathbf{y}), \lambda_t^2 I + \bar{\alpha}_t M_{t+1}^{-1}). \tag{32}$$

To match the prior in case of non-informative measurements, we set $\lambda_t^2 = 1 - \bar{\alpha}_t - \bar{\alpha}_t \rho_{t+1}^2 = 1 - \bar{\alpha}_t(1 + \rho_{t+1}^2)$.

For expressions with generalized DAPS prior (i.e., arbitrary value of $\eta$), see Appendix D.

## C. Comparison with TDS and MCGDiff

In this section, we write down the main components in an SMC algorithm, the target distributions and proposals, for Twisted Diffusion Sampler (TDS) (Wu et al., 2023) and Monte Carlo Guided Diffusion (MCGDiff) (Cardoso et al., 2024),

### C.1. TDS

**Target distributions**   TDS approximates the intractable likelihood $p(\mathbf{y}|\mathbf{x}_t)$ by $\hat{p}(\mathbf{y}|\mathbf{x}_t) \coloneqq p(\mathbf{y}|\mathbf{x}_0 = f_\theta(\mathbf{x}_t))$. The intermediate target distributions for TDS are hence

$$\gamma_t^{TDS}(\mathbf{x}_{t:T}) = \hat{p}(\mathbf{y}|\mathbf{x}_t)p(\mathbf{x}_T) \prod_{s=t}^{T-1} p_\theta(\mathbf{x}_s|\mathbf{x}_{s+1}), \tag{33}$$

where

$$p_\theta(\mathbf{x}_s|\mathbf{x}_{s+1}) = \mathcal{N}\left(\mathbf{x}_s|\mathbf{x}_{s+1} + \beta_{s+1}s_\theta(\mathbf{x}_{s+1}, s+1), \beta_{s+1}^2 I\right). \tag{34}$$

$$= \mathcal{N}\left(\mathbf{x}_s| \left(1 - \frac{\beta_{s+1}}{\sigma_{s+1}^2}\right)\mathbf{x}_{s+1} + \frac{\beta_{s+1}}{\sigma_{s+1}^2}f_\theta(\mathbf{x}_{s+1}), \beta_{t+1}\right), \tag{35}$$

and the last equality comes from rewriting the score as $(f_\theta(\mathbf{x}_{s+1}) - \mathbf{x}_{s+1})/\sigma_{s+1}^2$ according to Tweedie's formula.

**Proposal**   In TDS, they use the proposal

$$r_t^{TDS}(\mathbf{x}_t|\mathbf{x}_{t+1}, \mathbf{y}) = \mathcal{N}\left(\mathbf{x}_t|\mathbf{x}_{t+1} + \beta_{t+1}(s_\theta(\mathbf{x}_{t+1}, t+1) + \nabla_{\mathbf{x}_{t+1}} \log \hat{p}(\mathbf{y}|\mathbf{x}_{t+1})), \beta_{t+1}\right) \tag{36}$$

$$= \mathcal{N}\left(\mathbf{x}_t| \left(1 - \frac{\beta_{t+1}}{\sigma_{t+1}^2}\right)\mathbf{x}_{t+1} + \frac{\beta_{t+1}}{\sigma_{t+1}^2}f_\theta(\mathbf{x}_{t+1}) + \nabla_{\mathbf{x}_{t+1}} \log \hat{p}(\mathbf{y}|\mathbf{x}_{t+1}), \beta_{t+1}\right), \tag{37}$$

where again the last equality is from rewriting the score according to Tweedie's formula. This proposal is reminiscent of classifier guidance (Dhariwal & Nichol, 2021), where the unconditional score is combined with the gradient of some log likelihood, in this case the log likelihood $\log \hat{p}(\mathbf{y}|\mathbf{x}_{t+1})$ from the target distribution. As this likelihood relies on the reconstruction $f_\theta(\mathbf{x}_{t+1})$, computing the gradient of $\log \hat{p}(\mathbf{y}|\mathbf{x}_{t+1})$ with respect to $\mathbf{x}_{t+1}$ means differentiating through the reconstruction, which could be computationally very expensive. On the other hand, this proposal is not limited to the linear-Gaussian case, but works with any differentiable $\hat{p}(\mathbf{y}|\mathbf{x}_{t+1})$.

### C.2. MCGDiff

MCGDiff relies on the DDPM or Variance-preserving formulation of diffusion models when describing their method, and we will use it here. To discuss MCGDiff, we will work under their initial premises that $\mathbf{y} = \bar{\mathbf{x}} + \sigma_y\epsilon$, i.e., $A$ merely extracts the top coordinates of $\mathbf{x}$ (denoted $\bar{\mathbf{x}}$), and $\mathbf{y}$ is a potentially noisy observation of these coordinates[8]. The variable $\underline{\mathbf{x}}$ represents the completely unobserved coordinates. We also follow their presentation, starting with the assumption of a noise-free observation $\mathbf{y}$, then continuing to the noisy case.

#### C.2.1. NOISELESS CASE

**Target distributions**   In the noiseless case, they target for $t = 0$

$$\gamma_0(\mathbf{x}_{0:T}) = p_\theta(\mathbf{x}_T) \left[\prod_{t=1}^{T-1} p_\theta(\mathbf{x}_t|\mathbf{x}_{t+1})\right] \underbrace{p_\theta(\mathbf{y} \widehat{\underline{\mathbf{x}}_0})|\mathbf{x}_1)}_{\delta_\mathbf{y}(\bar{\mathbf{x}}_0)p_\theta(\underline{\mathbf{x}}_0|\mathbf{x}_1)}. \tag{38}$$

---

[8]For general $A$, they also make use of the singular value decomposition of $A$ to make a change of basis where the "new" $A$ has this property.

where $p_\theta$ is the regular diffusion model. By then defining a likelihood for $t = 0$ as $\bar{q}_{0|0}(\mathbf{y}|\overline{\mathbf{x}}_0) = \delta_\mathbf{y}(\overline{\mathbf{x}}_0)$ and $\bar{q}_{t|0}(\mathbf{y}|\overline{\mathbf{x}}_t) = \mathcal{N}(\sqrt{\bar{\alpha}_t}\mathbf{y}, (1 - \bar{\alpha}_t)I)$ for $t > 0$, they have a target on the general form

$$\gamma_t(\mathbf{x}_{t:T}) = \frac{\bar{q}_{t|0}(\mathbf{y}|\overline{\mathbf{x}}_t)}{\bar{q}_{t+1|0}(\mathbf{y}|\overline{\mathbf{x}}_{t+1})} p(\mathbf{x}_t|\mathbf{x}_{t+1})\gamma_{t+1}(\mathbf{x}_{t+1:T}). \tag{39}$$

**Proposal** For $0 < t < T$, they construct their proposal

$$r_t(\mathbf{x}_t|\mathbf{x}_{t+1}, \mathbf{y}) = \frac{p_\theta(\mathbf{x}_t|\mathbf{x}_{t+1})\bar{q}_{t|0}(\mathbf{y}|\overline{\mathbf{x}}_t)}{\int p_\theta(\mathbf{x}_t|\mathbf{x}_{t+1})\bar{q}_{t|0}(\mathbf{y}|\overline{\mathbf{x}}_t)d\mathbf{x}_t} = p_\theta(\underline{\mathbf{x}}_t|\mathbf{x}_{t+1})\frac{p_\theta(\overline{\mathbf{x}}_t|\mathbf{x}_{t+1})\bar{q}_{t|0}(\mathbf{y}|\overline{\mathbf{x}}_t)}{\int p_\theta(\overline{\mathbf{x}}_t|\mathbf{x}_{t+1})\bar{q}_{t|0}(\mathbf{y}|\overline{\mathbf{x}}_t)d\overline{\mathbf{x}}_t}, \tag{40}$$

and as everything is Gaussian (transition and likelihood), all terms can be computed exactly. In words, they use the "regular" diffusion model $p_\theta(\underline{\mathbf{x}}_t|\mathbf{x}_{t+1})$ for the unobserved coordinates, and for the observed coordinates, they also weigh in the likelihood in the update. For $t = 0$, they instead use

$$r_0(\mathbf{x}_0|\mathbf{x}_1, \mathbf{y}) = \delta_\mathbf{y}(\overline{\mathbf{x}}_0)p_\theta(\underline{\mathbf{x}}_0|\mathbf{x}_1) \tag{41}$$

**Weighting function** We have not written out the weighting function $\gamma_t(\mathbf{x}_{t:T})/(r_t(\mathbf{x}_t|\mathbf{x}_{t+1})\gamma_{t+1}(\mathbf{x}_{t+1:T}))$ for DDSMC and TDS, but we do for MCGDiff. With the target and proposal above, the weighting function becomes

$$w_t(\mathbf{x}_t, \mathbf{x}_{t+1}) = \underbrace{\frac{\int p_\theta(\mathbf{x}_t|\mathbf{x}_{t+1})\bar{q}_{t|0}(\mathbf{y}|\overline{\mathbf{x}}_t)d\mathbf{x}_t}{p_\theta(\mathbf{x}_t|\mathbf{x}_{t+1})\bar{q}_{t|0}(\mathbf{y}|\overline{\mathbf{x}}_t)}}_{1/q_t(\mathbf{x}_t|\mathbf{x}_{t+1})} \underbrace{\frac{\bar{q}_{t|0}(\mathbf{y}|\overline{\mathbf{x}}_t)}{\bar{q}_{t+1|0}(\mathbf{y}|\overline{\mathbf{x}}_{t+1})}p_\theta(\mathbf{x}_t|\mathbf{x}_{t+1})}_{\gamma_t(\mathbf{x}_{t:T})/\gamma_{t+1}(\mathbf{x}_{t+1:T})} = \tag{42}$$

$$= \frac{\int p_\theta(\mathbf{x}_t|\mathbf{x}_{t+1})\bar{q}_{t|0}(\mathbf{y}|\overline{\mathbf{x}}_t)d\mathbf{x}_t}{\bar{q}_{t+1|0}(\mathbf{y}|\overline{\mathbf{x}}_{t+1})}. \tag{43}$$

We do this as this weight does not actually depend on $\mathbf{x}_t$, and can hence be computed *before* sampling from the proposal. This opens up the possibility for a so called fully adapted particle filter, where one can sample exactly from the target and the importance weights are always $1/N$.

### C.2.2. NOISY CASE

In the noisy case, they assume that there exists a timestep $\tau$ such that $\sigma_y^2 = \frac{1-\bar{\alpha}_\tau}{\bar{\alpha}_\tau}$. They further define $\widetilde{\mathbf{y}}_\tau = \sqrt{\bar{\alpha}_\tau}\mathbf{y}$

They use as targets for $t \geq \tau$

$$\gamma_t(\mathbf{x}_{t:T}) = g_t(\mathbf{x}_t)p_\theta(\mathbf{x}_T)\prod_{s=t}^{T-1} p_\theta(\mathbf{x}_s|\mathbf{x}_{s+1}) \tag{44}$$

$$= \frac{g_t(\mathbf{x}_t)}{g_{t+1}(\mathbf{x}_{t+1})}p_\theta(\mathbf{x}_t|\mathbf{x}_{t+1})\gamma_{t+1}(\mathbf{x}_{t+1:T}) \tag{45}$$

with the potential

$$g_t(\mathbf{x}_t) = \mathcal{N}\left(\mathbf{x}_t|\sqrt{\bar{\alpha}_t}\mathbf{y}, (1 - (1 - \kappa)\frac{\bar{\alpha}_t}{\bar{\alpha}_\tau})I\right). \tag{46}$$

For $t = 0$, they target

$$\gamma_0(\mathbf{x}_0, \mathbf{x}_{\tau:T}) = p(\mathbf{y}|\overline{\mathbf{x}}_0)p_\theta(\mathbf{x}_0|\mathbf{x}_\tau)p_\theta(\mathbf{x}_T)\prod_{s=\tau}^{T-1} p(\mathbf{x}_s|\mathbf{x}_{s+1}) \tag{47}$$

$$= \frac{p(\mathbf{y}|\overline{\mathbf{x}}_0)p_\theta(\mathbf{x}_0|\mathbf{x}_\tau)}{g_\tau(\mathbf{x}_\tau)}\gamma_\tau(\mathbf{x}_{\tau:T}) \tag{48}$$

What this means is that they between $T$ and $\tau$ run their algorithm conditioning on $\mathbf{y}$, and then sample unconditionally between $\tau$ and $0$.

## D. Detailed Expressions for General DAPS Prior

We generalize the DAPS prior by tempering the "likelihood" $q(\mathbf{x}_{t+1}|\mathbf{x}_t)$, and denote

$$q_\eta(\mathbf{x}_t|\mathbf{x}_{t+1},\mathbf{x}_0) \propto q(\mathbf{x}_{t+1}|\mathbf{x}_t)^\eta q(\mathbf{x}_t|\mathbf{x}_0). \tag{49}$$

as in Equation (8).

### D.1. Variance Exploding

In the "Variance Exploding" setting that we have used throughout the paper, Equation (49) becomes

$$q_\eta(\mathbf{x}_t|\mathbf{x}_{t+1},\mathbf{x}_0) = \mathcal{N}\left(\frac{\beta_{t+1}}{\eta\sigma_t^2 + \beta_{t+1}}\mathbf{x}_0 + \frac{\eta\sigma_t^2}{\eta\sigma_t^2 + \beta_{t+1}}\mathbf{x}_{t+1}, \frac{\beta_{t+1}}{\eta\sigma_t^2 + \beta_{t+1}}I\right). \tag{50}$$

We then get our generalized backward kernel $p_\theta^\eta(\mathbf{x}_t|\mathbf{x}_{t+1})$ by replacing $\mathbf{x}_0$ with the reconstruction $f_\theta(\mathbf{x}_{t+1})$, i.e.,

$$p_\theta^\eta(\mathbf{x}_t|\mathbf{x}_{t+1}) = \mathcal{N}\left(\frac{\beta_{t+1}}{\eta\sigma_t^2 + \beta_{t+1}}f_\theta(\mathbf{x}_{t+1}) + \frac{\eta\sigma_t^2}{\eta\sigma_t^2 + \beta_{t+1}}\mathbf{x}_{t+1}, \frac{\beta_{t+1}\sigma_t^2}{\eta\sigma_t^2 + \beta_{t+1}}I\right) \tag{51}$$

For the proposal, the posterior $\tilde{p}(\mathbf{x}_0|\mathbf{x}_{t+1},\mathbf{y})$ remains as in Equation (16), but we use a generalization

$$r_t(\mathbf{x}_t|\mathbf{x}_{t+1},\mathbf{y}) = \int \tilde{q}_\eta(\mathbf{x}_t|\mathbf{x}_{t+1},\mathbf{x}_0)\tilde{p}(\mathbf{x}_0|\mathbf{x}_{t+1},\mathbf{y})d\mathbf{x}_0 \tag{52}$$

where $\tilde{q}_\eta(\mathbf{x}_t|\mathbf{x}_{t+1},\mathbf{x}_0)$ is on the form

$$q_\eta(\mathbf{x}_t|\mathbf{x}_{t+1},\mathbf{x}_0) = \mathcal{N}\left(\frac{\beta_{t+1}}{\eta\sigma_t^2 + \beta_{t+1}}\mathbf{x}_0 + \frac{\eta\sigma_t^2}{\eta\sigma_t^2 + \beta_{t+1}}\mathbf{x}_{t+1}, \lambda_t^2 I\right). \tag{53}$$

This gives the proposal

$$r_t(\mathbf{x}_t|\mathbf{x}_{t+1},\mathbf{y}) =$$

$$= \mathcal{N}\left(\frac{\beta_{t+1}}{\eta\sigma_t^2 + \beta_{t+1}}\mathbf{M}_{t+1}^{-1}\mathbf{b}_{t+1} + \frac{\eta\sigma_t^2}{\eta\sigma_t^2 + \beta_{t+1}}\mathbf{x}_{t+1}, \lambda_t^2 I + \left(\frac{\beta_{t+1}}{\eta\sigma_t^2 + \beta_{t+1}}\right)^2 \mathbf{M}_{t+1}^{-1}\right), \tag{54}$$

where we choose

$$\lambda_t^2 = \frac{\beta_{t+1}\sigma_t^2}{\eta\sigma_t^2 + \beta_{t+1}} - \left(\frac{\beta_{t+1}\rho_{t+1}}{\eta\sigma_t^2 + \beta_{t+1}}\right)^2 \tag{55}$$

so that the proposal matches the prior in case of non-informative measurements ($\sigma_y \to \infty$).

### D.2. DDPM/VE

Using the same techniques as for VE but replacing $q(\cdot)$ with its corresponding expressions, we get the transition

$$p_\theta^\eta(\mathbf{x}_t|\mathbf{x}_{t+1}) = \mathcal{N}\left(\frac{\sqrt{\bar{\alpha}_t}\beta_{t+1}f_\theta(\mathbf{x}_{t+1}) + \eta\sqrt{1-\beta_{t+1}}(1-\bar{\alpha}_t)\mathbf{x}_{t+1}}{\eta - \eta\beta_{t+1} - \eta\bar{\alpha}_{t+1} + \beta_{t+1}}, \frac{\beta_{t+1}(1-\bar{\alpha}_t)}{\eta - \eta\beta_{t+1} - \eta\bar{\alpha}_{t+1} + \beta_{t+1}}I\right), \tag{56}$$

and with the same posterior $\tilde{p}(\mathbf{x}_0|\mathbf{x}_t,\mathbf{x}_{t+1})$ and the same procedure as before, we get the proposal as

$$r_t(\mathbf{x}_t|\mathbf{x}_{t+1},\mathbf{y}) = \mathcal{N}\left(\frac{\sqrt{\bar{\alpha}_t}\beta_{t+1}\mathbf{M}_{t+1}^{-1}\mathbf{b}_{t+1} + \eta\sqrt{1-\beta_{t+1}}(1-\bar{\alpha}_t)\mathbf{x}_{t+1}}{\eta - \eta\beta_{t+1} - \eta\bar{\alpha}_{t+1} + \beta_{t+1}}, \lambda^2 I + \left(\frac{\sqrt{\bar{\alpha}_t}\beta_{t+1}}{\eta - \eta\beta_{t+1} - \eta\bar{\alpha}_{t+1} + \beta_{t+1}}\right)^2 \mathbf{M}_{t+1}^{-1}\right) \tag{57}$$

with

$$\lambda_t^2 = 1 - \bar{\alpha}_t - \left(\frac{\sqrt{\bar{\alpha}_t}\beta_{t+1}\rho_{t+1}}{\eta - \eta\beta_{t+1} - \eta\bar{\alpha}_{t+1} + \beta_{t+1}}\right)^2 \tag{58}$$

# E. Discrete Version

## E.1. Discrete Denoising Diffusion Probabilistic Models

DDSMC relies on the data $\mathbf{x}_0$ being a vector of continuous variables. For discrete data, there are a number of different formulations developed like (Austin et al., 2021; Campbell et al., 2022; Dieleman et al., 2022; Sun et al., 2023). In our work, we focus on the Discrete Denoising Diffusion Probabilistic Models (D3PM) model (Austin et al., 2021).

To introduce D3PM, we will denote by bold $\mathbf{x}$ a one-hot encoding of a single discrete variable. In this case, the forward noise process can be described as

$$q(\mathbf{x}_{t+1}|\mathbf{x}_t) = \mathbf{x}_t Q_{t+1}, \tag{59}$$

and as for the continuous case we have a closed form expression for $q(\mathbf{x}_t|\mathbf{x}_0)$ as

$$q(\mathbf{x}_t|\mathbf{x}_0) = \mathbf{x}_0 \bar{Q}_t \tag{60}$$

where $\bar{Q}_t = Q_1 \ldots Q_{t-1} Q_t$. The backward kernel $q(\mathbf{x}_t|\mathbf{x}_{t+1}, \mathbf{x}_0)$ is also known, and as we are in the discrete regime, the backward kernel can hence be parametrized by marginalizing over $\mathbf{x}_0$ as

$$p_\theta(\mathbf{x}_t|\mathbf{x}_{t+1}) = \sum_{x_0} q(\mathbf{x}_t|\mathbf{x}_{t+1}, \mathbf{x}_0)\tilde{p}_\theta(\mathbf{x}_0|\mathbf{x}_{t+1}), \tag{61}$$

where $\tilde{p}_\theta(\mathbf{x}_0|\mathbf{x}_{t+1})$ is a categorical distribution predicted by a neural network.

**Generalizing to $D$ Variables**  The reasoning above can be extended to general $D$-dimensional data by assuming a noise process where the noise is added independently to each ov the variables, meaning $q(\cdot)$ factorizes over these variables. Similarly, it is also assumed that the backward process $p_\theta(\cdot|\mathbf{x}_{t+1})$ factorizes over the different variables. A single forward pass through the neural networks hence produces probability vectors in $D$ different categorical distributions.

## E.2. Measurements

For the discrete setting where we have a single variable, we assume that we have a measurement $\mathbf{y}$ on the form

$$p(\mathbf{y}|\mathbf{x}_0) = \mathbf{x}_0 Q_y. \tag{62}$$

For the general $D$-dimensional case, we hence assume that we have $D$ measurements, one for each variable in $\mathbf{x}_0$, and that these are obtained independently from each other.

## E.3. D3SMC

In the discrete setting, it might be tempting to find an "optimal" proposal

$$r_t(\mathbf{x}_t|\mathbf{x}_{t+1}, \mathbf{y}) = \frac{p(\mathbf{y}|\mathbf{x}_t)p(\mathbf{x}_t|\mathbf{x}_{t+1})}{p(\mathbf{y}|\mathbf{x}_{t+1})}. \tag{63}$$

However, it is not possible to compute

$$p(\mathbf{y}|\mathbf{x}_t) = \sum_{\mathbf{x}_0} p(\mathbf{y}|\mathbf{x}_0)p(\mathbf{x}_0|\mathbf{x}_t) \tag{64}$$

in a single evaluation as it requires evaluations of the neural network prior for all possible values of $\mathbf{x}_t$. If only a single variable is updated at each denoising step, like in an order-agnostic autoregressive model (Uria et al., 2014), this could be feasible (Ekström Kelvinius & Lindsten, 2024), but for the $D$-dimensional case will require $d^D$ evaluations (where $d$ is the size of the state space of $\mathbf{x}_t$) for D3PM with uniform noise as all $D$ variables are updated each step. Hence, we have created a proposal which only requires a single forward pass at each sampling step in the general $D$-dimensional case.

Deriving the D3SMC algorithm is rather straight forward. In the derivation, we will continue using the notation with tilde. However, when we made Gaussian approximations around a reconstruction in the continuous case, we will in the discrete case use the categorical distribution predicted by our neural network, $\tilde{p}_\theta(\mathbf{x}_0|\mathbf{x}_{t+1})$.

**Target Distribution** In analogy with DDSMC, we write our target distribution $\gamma_t(\mathbf{x}_t)$ as

$$\gamma_t(\mathbf{x}_t) = \frac{\tilde{p}(\mathbf{y}|\mathbf{x}_t)}{\tilde{p}(\mathbf{y}|\mathbf{x}_{t+1})} p_\theta^0(\mathbf{x}_t|\mathbf{x}_{t+1}) \gamma_{t+1}(\mathbf{x}_{t+1}) \tag{65}$$

where $p_\theta^0(\mathbf{x}_t|\mathbf{x}_{t+1}) = \text{Categorical}(\mathbf{p} = \mathbf{x}_{0,t+1}^\theta \bar{Q}_t)$ with $\mathbf{x}_{0,t+1}^\theta$ being the predicted distribution over $\mathbf{x}_0$ by the D3PM model (i.e., $\tilde{p}_\theta(\mathbf{x}_0|\mathbf{x}_{t+1})$), and $\tilde{p}(\mathbf{y}|\mathbf{x}_t) = \sum_{\mathbf{x}_0} p(\mathbf{y}|\mathbf{x}_0) \tilde{p}_\theta(\mathbf{x}_0|\mathbf{x}_t)$. The distribution $p_\theta^0(\mathbf{x}_t|\mathbf{x}_{t+1})$ is a D3PM analogy to the DAPS kernel, as we predict $\mathbf{x}_0$ and then propagate this forward in time.

**Proposal Distribution** The discrete proposal starts, just as for the continuous case, with a posterior over $\mathbf{x}_0$,

$$\tilde{p}_\theta(\mathbf{x}_0|\mathbf{x}_{t+1}, \mathbf{y}) = \frac{p(\mathbf{y}|\mathbf{x}_0) \tilde{p}_\theta(\mathbf{x}_0|\mathbf{x}_{t+1})}{\sum_{\mathbf{x}_0} p(\mathbf{y}|\mathbf{x}_0) p_\theta(\mathbf{x}_0|\mathbf{x}_{t+1})} \tag{66}$$

$$= \frac{\mathbf{y} Q_y^T \odot \mathbf{x}_{0,t+1}^\theta}{Z_{t+1}}, \tag{67}$$

where $\odot$ means element-wise multiplication and $Z_t$ can be computed by summing the elements in the numerator. With this, we can obtain a proposal distribution in analogy with the continuous case

$$r_t(\mathbf{x}_t|\mathbf{x}_{t+1}, \mathbf{y}) = \sum_{\mathbf{x}_0} p(\mathbf{x}_t|\mathbf{x}_0) \tilde{p}(\mathbf{x}_0|\mathbf{x}_{t+1}, \mathbf{y}) =$$

$$= \sum_{\mathbf{x}_0} \mathbf{x}_0 \bar{Q}_t \left( \left( \frac{\mathbf{y} Q_y^T \odot \mathbf{x}_{0,t+1}^\theta}{Z_{t+1}} \right) (\mathbf{x}_0)^T \right). \tag{68}$$

# F. Protein Structure Completion

Levy et al. (2024) provides a protein toy example where $\mathbf{x} \in \mathbb{R}^{N \times 4 \times 3}$ is the coordinates of the backbone. They then propose a general observation model

$$\mathbf{y} = A\mathbf{x}_0 + \eta, \quad \eta \sim \mathcal{N}(0, \boldsymbol{\Sigma}). \tag{69}$$

However, the diffusion prior is based on Chroma[9] (Ingraham et al., 2023), which models the diffusion as

$$d\mathbf{x} = -\frac{1}{2}\mathbf{x}\beta_t dt + \sqrt{\beta_t} R d\mathbf{w}, \tag{70}$$

which is the regular continuous-time formulations of DDPM/VP process but with *correlated* noise. Therefore, we can go to a new space $\mathbf{z}$ defined as

$$\mathbf{z} = R^{-1}\mathbf{x}. \tag{71}$$

Now, in this space, the diffusion process follows the standard DDPM/VP diffusion process and we can at any point convert between the two spaces with

$$\mathbf{x}_t = R\mathbf{z}_t, \tag{72}$$

or in other words, at any point

$$\mathbf{z}_t|\mathbf{z}_0 \sim \mathcal{N}(\mathbf{z}_t|\sqrt{\bar{\alpha}_t}\mathbf{z}_0, (1 - \bar{\alpha}_t)I). \tag{73}$$

Going back to the measurement equation, we can write it in terms of the whitened space $\mathbf{z}$ as

$$\mathbf{y} = AR\mathbf{z} + \eta. \tag{74}$$

---

[9]This includes using pretrained weight parameters, trained by GENERATE BIOMEDICINES, INC. who are the copyright holders of these weights. The weights have been used with permission according to the license at https://chroma-weights.generatebiomedicines.com/

Now, Levy et al. (2024) make an assumption which is equivalent to the measurement

$$\mathbf{y} = AR(\mathbf{z}_0 + \tilde{\eta}), \quad \tilde{\eta} \sim \mathcal{N}(0, \sigma^2 I_{d_z}), \tag{75}$$

where $d_z$ is the dimension of $\mathbf{z}$ (which is the same as $d_x$). This is equivalent to the previous measurement equation with $\Sigma = \sigma^2 U S S^T U^T$, where $U$ and $S$ are from the SVD of the matrix $AR = USV^T$.

Now, to run DDSMC we can rewrite the measurement equation using the aforementioned SVD as

$$\mathbf{y} = USV^T \mathbf{z}_0 + USV^T \tilde{\eta} \iff U^T \mathbf{y} = SV^T \mathbf{z}_0 + SV^T \tilde{\eta} \tag{76}$$

and with a change of basis as $\mathbf{y}' = U^T \mathbf{y}$, $\mathbf{z}' = V^T \mathbf{z}$, $A' = S$ and $\tilde{\eta}' = SV^T \sim \mathcal{N}(0, \sigma^2 SV^T V S^T) = \mathcal{N}(0, \sigma^2 S S^T)$ the measurement equation becomes

$$\mathbf{y}' = A' \mathbf{z}_0' + \tilde{\eta}', \quad \tilde{\eta}' \sim \mathcal{N}(0, \sigma^2 S S^T). \tag{77}$$

We can now run DDSMC in this new basis with diagonal matrix $A'$. One should note here that the covariance do not necessarily have constant diagonal entries, which will slightly change the expressions for DDSMC: instead of relying on a single $\sigma_y$, we have to replace this in the detailed expressions with $s_i \sigma$ for the dimension $i \leq d_y$, where $s_i$ is the corresponding singular value $S_{ii}$.

# G. Additional Experimental Details and Results

## G.1. Gaussian Mixture Model

### G.1.1. PRIOR AND POSTERIOR

For the GMM experiments, we tried following MCGDiff as closely as possible, and therefore used and modified code from the MCGDiff repository[10]. We briefly outline the setup here, but refer to their paper for the full details.

The data prior is a Gaussian Mixture with 25 components, each having unit covariances. Under the DDPM/VP model (see Appendix B), the intermediate marginal distributions $q(\mathbf{x}_t) = \int q(\mathbf{x}_t|\mathbf{x}_0)q(\mathbf{x}_0)d\mathbf{x}_0$ will also be a mixture of Gaussians, and it is hence possible to compute $\nabla_{\mathbf{x}_t} \log q(\mathbf{x}_t)$ exactly.

The sequence of $\beta_t$ is a linearly decreasing sequence between $0.02$ and $10^{-4}$.

The mixture weights and the measurement model $(A, \sigma_y)$ are randomly generated, and a measurement is then drawn by sampling first $\mathbf{x}_0^* \sim q(\mathbf{x}_0)$, then $\epsilon \sim \mathcal{N}(0, I)$, and finally setting $y = A\mathbf{x}^* + \sigma_y \epsilon$. From this measurement, it is possible to compute the posterior exactly, and 10k samples were drawn from the posterior. Additionally, 10k samples were generated by each of the algorithms, and the Sliced Wasserstein Distance between the exact posterior samples and the SMC samples were computed using the Python Optimal Transport (POT) package[11] (Flamary et al., 2021)

### G.1.2. IMPLEMENTATION DETAILS

For the three SMC methods, we used 256 particles for all algorithms, and 20 DDIM timesteps (Song et al., 2021a). In this case, the timesteps were chosen using the method described in MCGDiff.

For DDSMC and DAPS, we used the DDIM ODE solver (see equation below), and solve the ODE for the "remaining steps" in the diffusion process. This means that, at diffusion time $t$, we start with a sample $\mathbf{x}_{t'} = \mathbf{x}_t$ and then update this sample according to the update rule

$$\mathbf{x}_{t'-1} = \sqrt{\frac{\bar{\alpha}_{t'-1}}{\bar{\alpha}_{t'}}} \mathbf{x}_{t'} + \left( (1 - \bar{\alpha}_{t'}) \sqrt{\frac{\bar{\alpha}_{t'-1}}{\bar{\alpha}_{t'}}} - \sqrt{(1 - \bar{\alpha}_{t'-1})(1 - \bar{\alpha}_{t'})} \right) \nabla_{\mathbf{x}_{t'}} q(\mathbf{x}_{t'}). \tag{78}$$

for $t' = t, t-1, \ldots, 1$.

**DDSMC** We use $\rho_t^2 = (1 - \bar{\alpha}_t)/\sqrt{2}$

---

[10]https://github.com/gabrielvc/mcg_diff
[11]https://pythonot.github.io/index.html

*Table 4.* Comparison of DDRM with 20 or 1 000 steps in the GMM setting.

| $d_x$ | $d_y$ | DDRM-20 | DDRM-1 000 |
|---|---|---|---|
| 8 | 1 | $5.22 \pm 0.92$ | $3.83 \pm 1.05$ |
|   | 2 | $4.58 \pm 1.09$ | $2.25 \pm 0.92$ |
|   | 4 | $2.05 \pm 0.85$ | $0.55 \pm 0.29$ |
| 80 | 1 | $6.32 \pm 1.24$ | $5.19 \pm 1.07$ |
|   | 2 | $7.60 \pm 0.90$ | $5.62 \pm 1.09$ |
|   | 4 | $7.74 \pm 0.84$ | $4.95 \pm 1.25$ |
| 800 | 1 | $7.02 \pm 1.14$ | $7.15 \pm 1.11$ |
|   | 2 | $8.29 \pm 0.86$ | $8.21 \pm 0.88$ |
|   | 4 | $9.07 \pm 0.73$ | $8.66 \pm 0.87$ |

**MCGDiff** We used code from the official implementation[12]. As per their paper, we used $\kappa = 10^{-2}$.

**TDS** We implemented the method ourselves according to their Algorithm 1, making the adaptions for the DDPM/VP setting (also described in their Appendix A.1).

**DAPS** We implemented the method ourselves. As we are in the linear-Gaussian setting, we replaced their approximate sampling of $\tilde{p}(\mathbf{x}_0|\mathbf{x}_t, \mathbf{y})$ with the exact expression (given by our Equations (16) and (17)). We used the same schedule for $r_t$ as for our $\rho_t$.

**DDRM** We used code from the official repository[13], and used the same hyper-parameters $\eta$ and $\eta_b$ as in their paper. We used 1 000 timesteps to avoid discretization error, but also tested 20 steps as it is argued in their paper that more is not necessarily better. This performed worse for $d_x = 8$ and 80, and similar for $d_x = 800$, see Table 4.

**DCPS** We used code from the official repository [14]. We used $L = 3$, $K = 2$, and $\gamma = 10^{-2}$ as specified in their paper. We used 1 000 timesteps.

### G.1.3. COMPLEXITY AND NUMBER OF FUNCTION EVALUATIONS

As DDSMC relies on multiple particles in parallel, it will inevitably require a larger computational effort. DDSMC-Tweedie uses one score evaluation per particle per diffusion step, thus requiring $N$ times more score evaluations per step compared to a method like DDRM (with $N$ being the number of particles). This is the same effort as MCGDiff, and slightly less than TDS which requires differentiating the score function. DDSMC-ODE and DAPS uses additional steps for the reconstruction function, and DDSMC-ODE hence requires $N$ times more score evaluations per diffusion step than DAPS, which in turn requires "number of ODE steps" times more score evaluations than a method like DDRM. In the case of the GMM setting, we used 20 diffusion steps, and reconstruction required $20, 19, \ldots, 1$ steps, meaning in total it required $\sim 10$ times more score evaluations just for reconstruction. As noted in the paper, using DDSMC-Tweedie already outperforms all other methods (including MCGDiff, which has the same computational footprint). We investigate further the spending of the computational effort in DDSMC-ODE in Appendix G.1.5.

### G.1.4. DDSMC-ODE WITH SINGLE PARTICLE (COMPARISON WITH DAPS)

As the proposal of DDSMC-ODE with $\eta = 0$ essentially corresponds to DAPS (if instead using $\lambda^2 = \sigma^2$ in Equation (19)), running this setup with a single particle would become essentially equivalent to DAPS. We verify that this also holds empirically in Table 5, where we present the numbers for DDSMC-ODE with a single particle. Indeed, the numbers for $\eta = 0$ are more or less identical.

---

[12]https://github.com/gabrielvc/mcg_diff
[13]https://github.com/bahjat-kawar/ddrm
[14]https://github.com/Badr-MOUFAD/dcps/

*Table 5.* Results on the Gaussian mixture model experiment, using DDSMC-ODE with 20 ODE steps and $N = 1$ particles, i.e., sampling from the proposal. The numbers for $\eta = 0$ are more or less identical to DAPS, verifying that our proposal and their method are essentially equivalent also in practice.

| $d_x$ | $d_y$ | $\eta = 0.0$ | $\eta = 0.5$ | $\eta = 1.0$ |
|---|---|---|---|---|
| 8 | 1 | $5.62 \pm 0.90$ | $5.85 \pm 0.93$ | $6.26 \pm 0.96$ |
| | 2 | $5.93 \pm 1.16$ | $6.33 \pm 1.07$ | $6.92 \pm 1.00$ |
| | 4 | $4.88 \pm 1.34$ | $5.42 \pm 1.25$ | $6.16 \pm 1.16$ |
| 80 | 1 | $6.84 \pm 1.16$ | $7.15 \pm 1.11$ | $7.49 \pm 1.10$ |
| | 2 | $8.50 \pm 0.92$ | $8.87 \pm 0.90$ | $9.22 \pm 0.90$ |
| | 4 | $9.05 \pm 0.75$ | $9.42 \pm 0.71$ | $9.78 \pm 0.68$ |
| 800 | 1 | $7.03 \pm 1.20$ | $7.13 \pm 1.19$ | $7.26 \pm 1.17$ |
| | 2 | $8.30 \pm 1.01$ | $8.41 \pm 1.02$ | $8.57 \pm 1.01$ |
| | 4 | $9.20 \pm 0.83$ | $9.33 \pm 0.84$ | $9.50 \pm 0.83$ |

*Table 6.* Results on the Gaussian mixture model experiment when using DDSMC-ODE, $\eta = 0$, with fewer ODE steps and/or fewer particles N. *This setup requires as many score evaluations as DDSMC-Tweedie with $N = 256$ particles.

| | | ODE STEPS = 3 | | ODE STEPS = 7 | | ODE STEPS = 20 | | |
|---|---|---|---|---|---|---|---|---|
| $d_x$ | $d_y$ | $N = 90^*$ | $N = 256$ | $N = 40^*$ | $N = 256$ | $N = 12$ | $N = 25^*$ | $N = 256$ |
| 8 | 1 | $1.89 \pm 0.57$ | $1.79 \pm 0.52$ | $1.68 \pm 0.50$ | $1.49 \pm 0.40$ | $0.98 \pm 0.29$ | $0.94 \pm 0.20$ | $1.15 \pm 0.24$ |
| | 2 | $0.63 \pm 0.30$ | $0.63 \pm 0.30$ | $0.55 \pm 0.27$ | $0.53 \pm 0.26$ | $0.57 \pm 0.25$ | $0.48 \pm 0.18$ | $0.56 \pm 0.23$ |
| | 4 | $0.16 \pm 0.01$ | $0.16 \pm 0.01$ | $0.17 \pm 0.02$ | $0.17 \pm 0.04$ | $0.18 \pm 0.04$ | $0.19 \pm 0.07$ | $0.21 \pm 0.10$ |
| 80 | 1 | $1.22 \pm 0.36$ | $1.20 \pm 0.36$ | $1.14 \pm 0.34$ | $1.11 \pm 0.33$ | $0.71 \pm 0.16$ | $0.70 \pm 0.17$ | $0.69 \pm 0.17$ |
| | 2 | $1.15 \pm 0.42$ | $1.08 \pm 0.41$ | $1.09 \pm 0.48$ | $0.97 \pm 0.42$ | $0.88 \pm 0.57$ | $0.70 \pm 0.39$ | $0.69 \pm 0.36$ |
| | 4 | $0.53 \pm 0.28$ | $0.55 \pm 0.30$ | $0.59 \pm 0.29$ | $0.55 \pm 0.30$ | $0.71 \pm 0.45$ | $0.40 \pm 0.19$ | $0.41 \pm 0.23$ |
| 800 | 1 | $1.69 \pm 0.54$ | $1.66 \pm 0.53$ | $1.65 \pm 0.53$ | $1.56 \pm 0.49$ | $1.56 \pm 0.49$ | $1.39 \pm 0.47$ | $1.37 \pm 0.43$ |
| | 2 | $1.24 \pm 0.64$ | $1.08 \pm 0.51$ | $1.42 \pm 0.95$ | $1.07 \pm 0.45$ | $1.64 \pm 0.86$ | $1.23 \pm 0.78$ | $0.81 \pm 0.40$ |
| | 4 | $0.38 \pm 0.27$ | $0.25 \pm 0.08$ | $0.50 \pm 0.40$ | $0.24 \pm 0.08$ | $1.18 \pm 0.82$ | $0.70 \pm 0.60$ | $0.23 \pm 0.06$ |

*Table 7.* Results on the Gaussian mixture model experiment when using DDSMC-ODE, $\eta = 0.5$, with fewer ODE steps and/or fewer particles N. *This setup requires as many score evaluations as DDSMC-Tweedie with $N = 256$ particles.

| | | ODE STEPS = 3 | | ODE STEPS = 7 | | ODE STEPS = 20 | | |
|---|---|---|---|---|---|---|---|---|
| $d_x$ | $d_y$ | $N = 90^*$ | $N = 256$ | $N = 40^*$ | $N = 256$ | $N = 12$ | $N = 25^*$ | $N = 256$ |
| 8 | 1 | $1.92 \pm 0.73$ | $1.84 \pm 0.64$ | $1.49 \pm 0.52$ | $1.33 \pm 0.41$ | $0.70 \pm 0.25$ | $0.69 \pm 0.25$ | $0.88 \pm 0.33$ |
| | 2 | $0.62 \pm 0.36$ | $0.61 \pm 0.35$ | $0.44 \pm 0.29$ | $0.45 \pm 0.29$ | $0.44 \pm 0.23$ | $0.30 \pm 0.20$ | $0.34 \pm 0.20$ |
| | 4 | $0.08 \pm 0.03$ | $0.09 \pm 0.04$ | $0.05 \pm 0.02$ | $0.05 \pm 0.01$ | $0.07 \pm 0.05$ | $0.07 \pm 0.06$ | $0.09 \pm 0.08$ |
| 80 | 1 | $1.01 \pm 0.33$ | $1.01 \pm 0.34$ | $0.96 \pm 0.29$ | $0.92 \pm 0.30$ | $0.64 \pm 0.14$ | $0.53 \pm 0.14$ | $0.50 \pm 0.16$ |
| | 2 | $0.91 \pm 0.39$ | $0.85 \pm 0.40$ | $0.95 \pm 0.43$ | $0.77 \pm 0.36$ | $0.91 \pm 0.50$ | $0.57 \pm 0.27$ | $0.37 \pm 0.20$ |
| | 4 | $0.38 \pm 0.25$ | $0.38 \pm 0.27$ | $0.43 \pm 0.23$ | $0.36 \pm 0.24$ | $0.58 \pm 0.33$ | $0.31 \pm 0.14$ | $0.21 \pm 0.13$ |
| 800 | 1 | $1.81 \pm 0.63$ | $1.76 \pm 0.60$ | $2.01 \pm 0.65$ | $1.82 \pm 0.57$ | $2.20 \pm 0.63$ | $1.92 \pm 0.59$ | $1.68 \pm 0.52$ |
| | 2 | $1.51 \pm 0.97$ | $1.27 \pm 0.78$ | $1.96 \pm 1.11$ | $1.34 \pm 0.77$ | $2.38 \pm 1.00$ | $1.94 \pm 0.97$ | $1.17 \pm 0.65$ |
| | 4 | $0.49 \pm 0.33$ | $0.33 \pm 0.15$ | $0.77 \pm 0.45$ | $0.38 \pm 0.16$ | $1.70 \pm 0.81$ | $1.10 \pm 0.66$ | $0.47 \pm 0.22$ |

### G.1.5. DDSMC-ODE WITH FEWER PARTICLES AND/OR ODE STEPS

As DDSMC-ODE requires additional score evaluations when solving the PF-ODE, we present some results in Tables 6 to 8 where we have used fewer particles and/or fewer ODE steps. Notice that the number of ODE steps is the *maximum* number of ODE steps, and if there are fewer steps left in the (outer) diffusion process, this lower number is used as number of ODE steps that iteration. Hence, as an example, with the maximum number of ODE steps being 20 and the number of diffusion steps also being 20, the number of ODE steps used in the different iterations would be $20, 19, \ldots, 1$, and

*Table 8.* Results on the Gaussian mixture model experiment when using DDSMC-ODE, $\eta = 1$, with fewer ODE steps and/or fewer particles N. *This setup requires as many score evaluations as DDSMC-Tweedie with $N = 256$ particles.

| | | ODE STEPS = 3 | | ODE STEPS = 7 | | ODE STEPS = 20 | | |
|---|---|---|---|---|---|---|---|---|
| $d_x$ | $d_y$ | $N = 90^*$ | $N = 256$ | $N = 40^*$ | $N = 256$ | $N = 12$ | $N = 25^*$ | $N = 256$ |
| | 1 | $2.07 \pm 0.81$ | $1.99 \pm 0.71$ | $1.44 \pm 0.51$ | $1.30 \pm 0.38$ | $0.85 \pm 0.25$ | $0.76 \pm 0.25$ | $1.01 \pm 0.39$ |
| 8 | 2 | $0.75 \pm 0.39$ | $0.69 \pm 0.38$ | $0.57 \pm 0.27$ | $0.50 \pm 0.29$ | $0.70 \pm 0.22$ | $0.48 \pm 0.19$ | $0.39 \pm 0.22$ |
| | 4 | $0.14 \pm 0.06$ | $0.13 \pm 0.06$ | $0.19 \pm 0.03$ | $0.14 \pm 0.02$ | $0.29 \pm 0.05$ | $0.23 \pm 0.05$ | $0.17 \pm 0.06$ |
| | 1 | $1.14 \pm 0.24$ | $1.10 \pm 0.23$ | $1.17 \pm 0.21$ | $1.06 \pm 0.21$ | $1.11 \pm 0.14$ | $0.91 \pm 0.11$ | $0.77 \pm 0.12$ |
| 80 | 2 | $1.08 \pm 0.36$ | $1.02 \pm 0.34$ | $1.18 \pm 0.35$ | $0.97 \pm 0.30$ | $1.39 \pm 0.46$ | $0.99 \pm 0.26$ | $0.69 \pm 0.15$ |
| | 4 | $0.69 \pm 0.16$ | $0.63 \pm 0.17$ | $0.79 \pm 0.18$ | $0.67 \pm 0.15$ | $1.01 \pm 0.27$ | $0.75 \pm 0.15$ | $0.54 \pm 0.04$ |
| | 1 | $2.17 \pm 0.62$ | $2.09 \pm 0.58$ | $2.44 \pm 0.64$ | $2.21 \pm 0.55$ | $2.67 \pm 0.64$ | $2.36 \pm 0.60$ | $2.09 \pm 0.51$ |
| 800 | 2 | $2.05 \pm 0.95$ | $1.81 \pm 0.81$ | $2.56 \pm 1.08$ | $1.90 \pm 0.79$ | $2.96 \pm 0.99$ | $2.52 \pm 0.96$ | $1.76 \pm 0.67$ |
| | 4 | $1.18 \pm 0.48$ | $1.04 \pm 0.37$ | $1.50 \pm 0.54$ | $1.15 \pm 0.39$ | $2.38 \pm 0.80$ | $1.81 \pm 0.70$ | $1.24 \pm 0.47$ |

DDSMC-ODE would in this case use $\sim 10\times$ more score evaluations in total. Therefore, using $N = 25$ would corresponding to approximately the same number of score evaluations as DDSMC-Tweedie.

In the tables, a general trend seems to be that more ODE steps are more important than more particles under a compute budget, as 20 ODE steps with 25 particles seems to perform better than 3 and 7 ODE steps with 90 and 40 particles, respectively. It also in general performs better than DDSMC-Tweedie (which in principle uses a single ODE step). This could lead to the conclusion that one should use as many steps possible, and then add more particles up to the desired computational budget. However, running a *single* particle does not work well, and it hence seems like both aspects (more ODE-steps and more particles) are needed for optimal performance.

It should be noted that we have used number of score evaluations as measure of computational effort, and as mentioned above in the discussion on complexity, since particles are run in parallel, adding more particles might not have the same effect on computational *time* as adding more steps (if keeping the number of score evaluations equal). As more particles also improves performance, exactly how to tune the number of steps and particles can depend also on possibility for parallelization.

### G.1.6. ADDITIONAL QUALITATIVE RESULTS

In Figures 6 to 8 we present additional qualitative results to compare DDSMC with different $\eta$ and reconstruction functions. In Figures 9 to 11, we present extended qualitative comparisons with other methods.

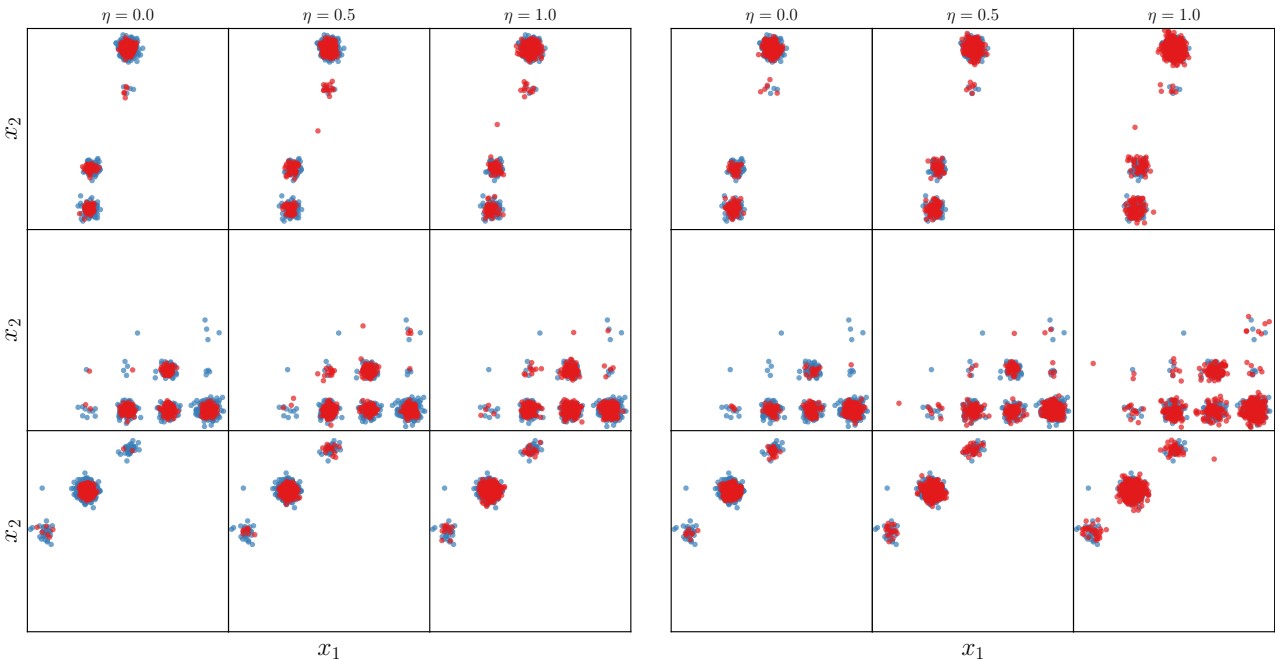

*Figure 6.* Extended qualitative results on GMM experiments, comparing DDSMC with Tweedie (three left-most figures) or PF-ODE as reconstruction, $d_x = 8$, $d_y = 1$.

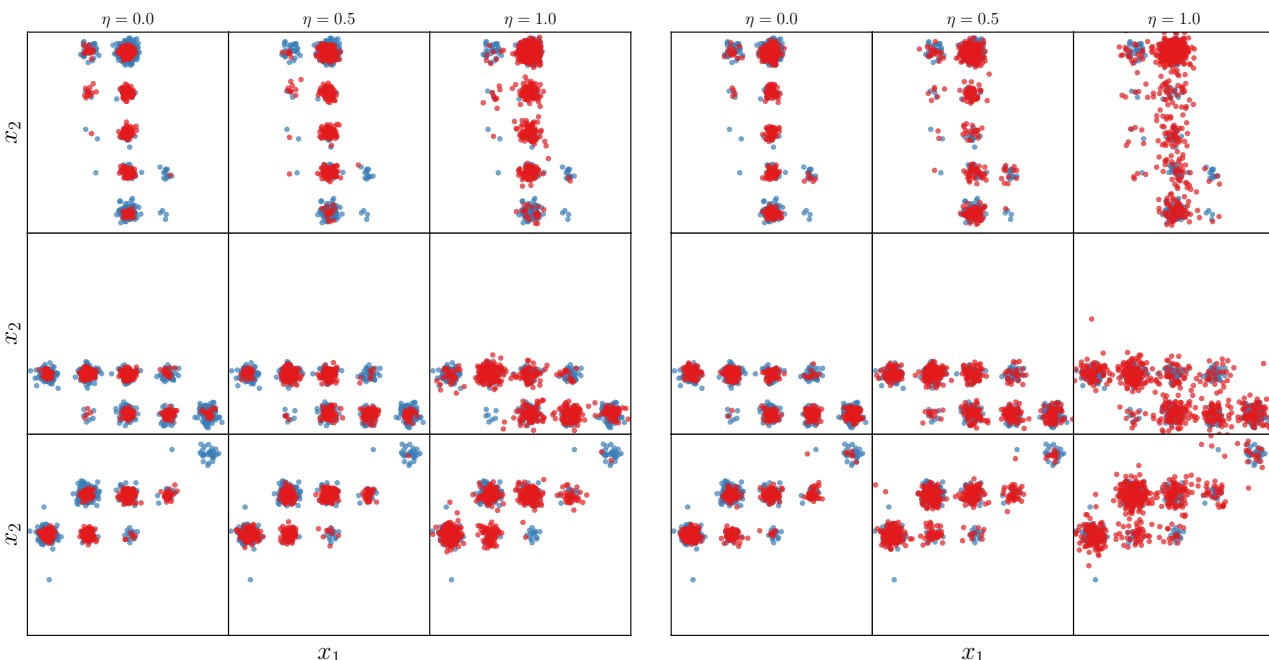

*Figure 7.* Extended qualitative results on GMM experiments, comparing DDSMC with Tweedie (three left-most figures) or PF-ODE as reconstruction, $d_x = 80$, $d_y = 1$.

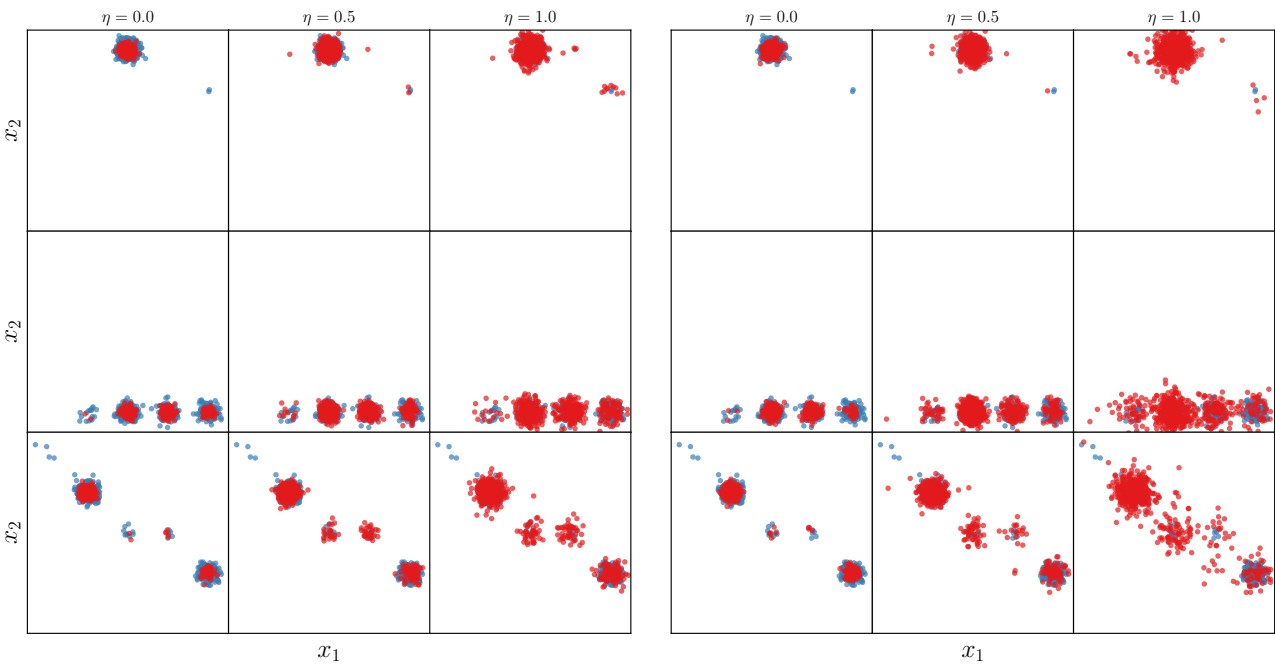

*Figure 8.* Extended qualitative results on GMM experiments, comparing DDSMC with Tweedie (three left-most figures) or PF-ODE as reconstruction, $d_x = 800$, $d_y = 1$.

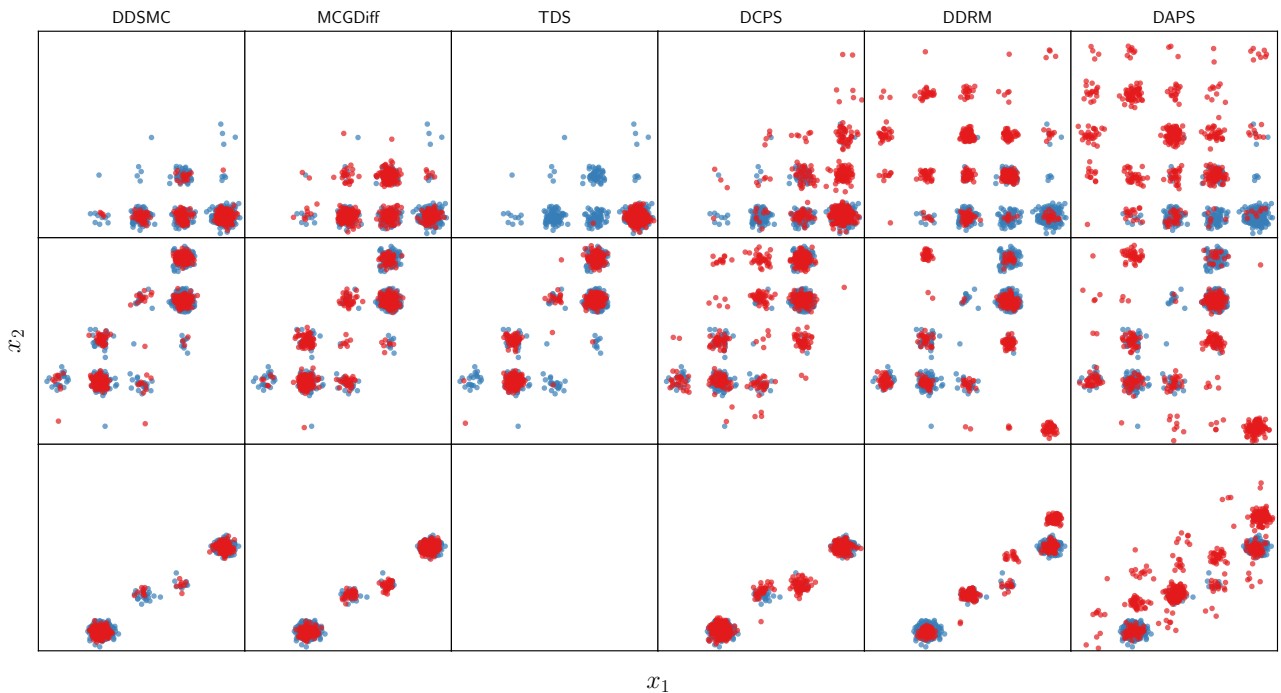

*Figure 9.* Extended qualitative results on GMM experiments, $d_x = 8$, $d_y = 1$. DDSMC using PF-ODE as reconstruction and $\eta = 0$.

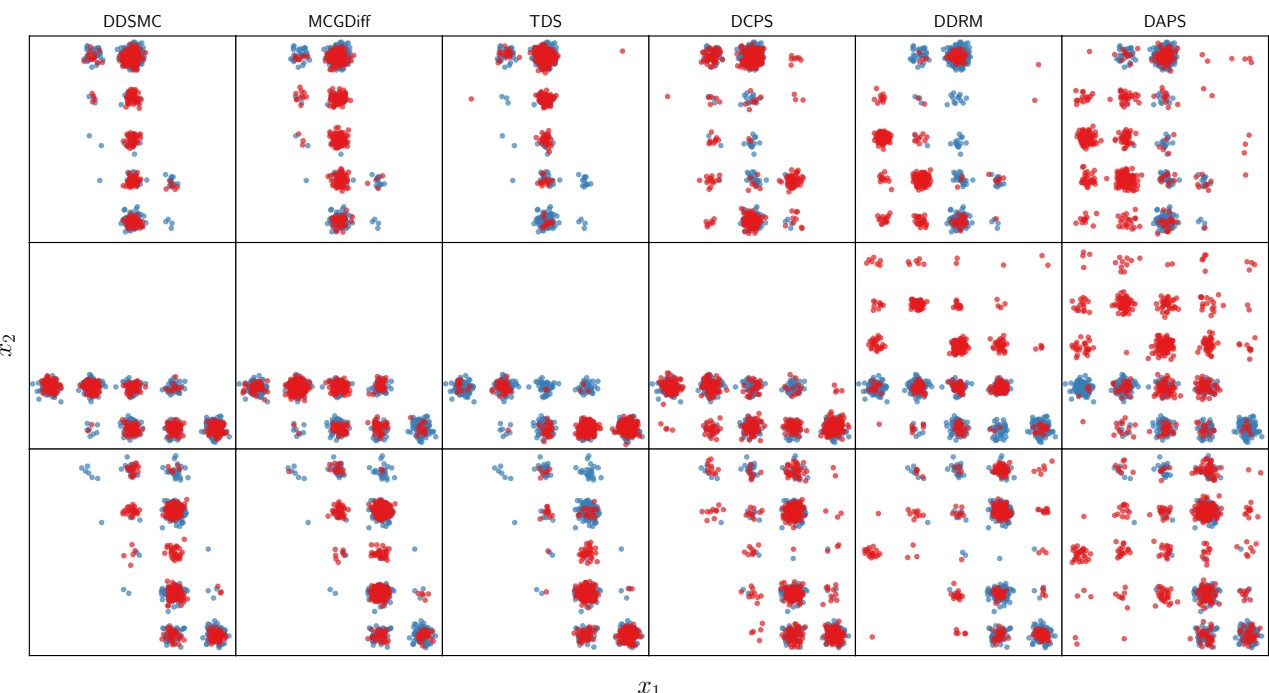

*Figure 10.* Extended qualitative results on GMM experiments, $d_x = 80$, $d_y = 1$. DDSMC using PF-ODE as reconstruction and $\eta = 0$.

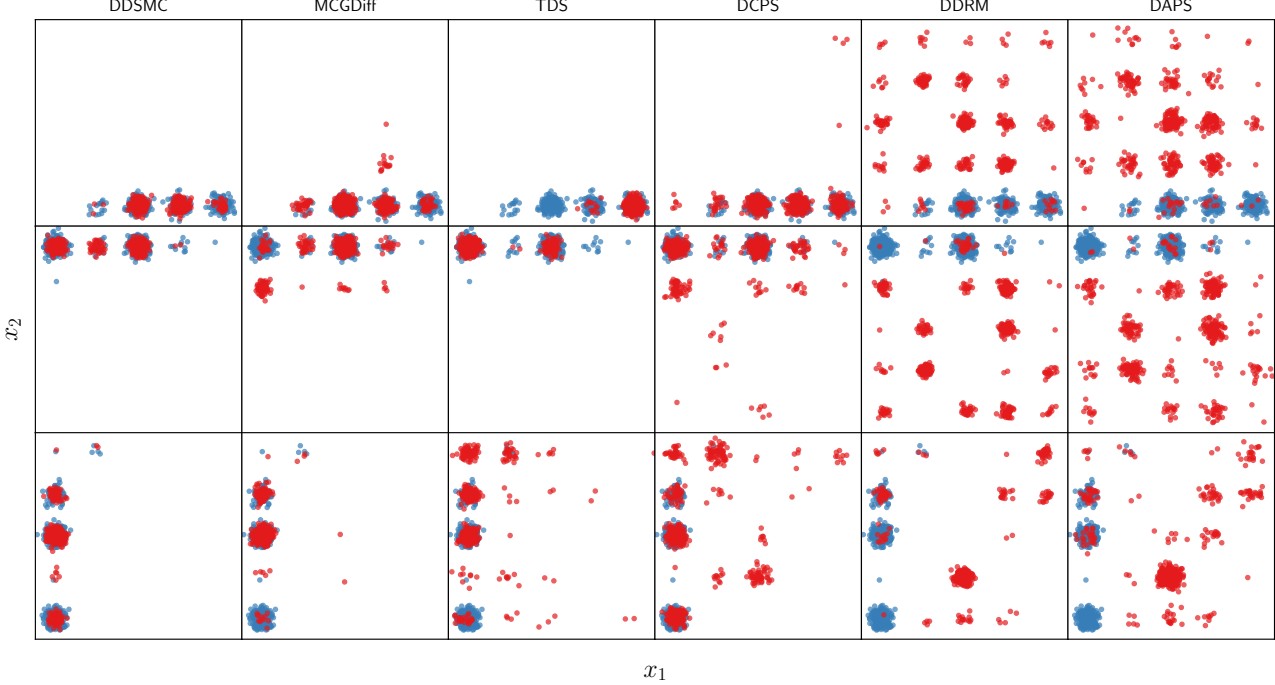

*Figure 11.* Extended qualitative results on GMM experiments, $d_x = 800$, $d_y = 1$. DDSMC using PF-ODE as reconstruction and $\eta = 0$.

## G.2. Image restoration

We perform the image restoration on the FFHQ dataset (Karras et al., 2019), downsampled from $1024 \times 1024$ to $256 \times 256$ pixels.

### G.2.1. DDSMC IMPLEMENTATION DETAILS AND HYPERPARAMETERS

For the image restoration tasks, we used $N = 5$ particles. We also follow DAPS and take on the view of diffusion as a continuous process, and the backward process as a discretization of this process at timesteps $t_1, t_2, \ldots, t_T$. With this, $t$ and $t + 1$ in Algorithm 1 are replaced by $t_i$ and $t_{i+1}$, respectively.

**Time and $\sigma_t$ Schedule**  We follow DAPS which uses a noise-schedule $\sigma_{t_i} = t_i$ where the timepoints $t_i$ $(i = 1, \ldots, T)$ are chosen as

$$t_i = \left( t_{\max}^{1/7} + \frac{T - i}{T - 1} \left( t_{\min}^{1/7} - t_{\max}^{1/7} \right) \right)^7 \tag{79}$$

We used $t_{\max} = 100$, $t_{\min} = 0.1$ and $T = 200$

**$\rho_t$ schedule**  In their public implementation[15], DAPS uses $\rho_t = \sigma$. However, we find that for when using the exact solution $\tilde{p}(\mathbf{x}_0|\mathbf{x}_{t+1}, \mathbf{y})$, these values are too high. We therefore lower $\rho_t$ by using a similar schedule as for $\sigma_t$,

$$\rho_{t_i} = \left( \rho_{\max}^{1/7} + \frac{T - i}{T - 1} \left( \rho_{\min}^{1/7} - \rho_{\max}^{1/7} \right) \right)^7 \tag{80}$$

but with with $\rho_{\max} = 50$ and $\rho_{\min} = 0.03$.

**PF-ODE**  Again, we followed DAPS and used an Euler solver to solve the PF-ODE by Karras et al. (2022), using 5 steps with $\sigma$ according to the same procedure as for the outer diffusion process (i.e., Equation (79)), at each step $t_i$ starting with $t_{\max} = t_i$ and ending with $t_{\min} = 0.01$.

### G.2.2. IMPLEMENTATION OF OTHER METHODS

For the other methods, we used the DCPS codebase [16] as basis where we implemented DDSMC and transferred DAPS.

**DAPS**  We used the hyperparameters as in their public repository[17].

**DCPS**  We used the settings from their public repository, running the algorithm for 300 steps as that was what was specified in their repository.

**DDRM**  Just as for DCPS, we used 300 steps. We also tried 20 steps as stated in the original paper, but found that this worked worse (see Tables 9 and 10).

**MCGDiff**  We used 32 particles.

### G.2.3. FULL QUANTITATIVE RESULTS

In Table 9 we present the LPIPS values with standard deviations, and include DDRM with both 20 and 300 steps. In Table 10 we present PSNR values.

### G.2.4. FFHQ IMAGES ATTRIBUTIONS

In Table 11, attribution information for the images from FFHQ displayed in this paper are provided.

---

[15]https://github.com/zhangbingliang2019/DAPS/
[16]https://github.com/Badr-MOUFAD/dcps/
[17]https://github.com/zhangbingliang2019/DAPS

*Table 9.* LPIPS results on FFHQ experiments. The noise level is $\sigma_y = 0.05$, and the tasks are inpainting a box in the **middle**, outpainting right **half** of the image, and **s**uper-**r**esolution $4\times$. Numbers are averages and standard deviations over 1k images. DDRM-20 and DDRM-300 means DDRM with 20 and 300 steps respectively. **Lower is better.**

|  |  | MIDDLE | HALF | SR4 |
|---|---|---|---|---|
|  | DDRM-20 | $0.08 \pm 0.02$ | $0.29 \pm 0.05$ | $0.2 \pm 0.05$ |
|  | DDRM-300 | $0.04 \pm 0.01$ | $0.25 \pm 0.05$ | $0.2 \pm 0.05$ |
|  | DCPS | $0.03 \pm 0.01$ | $0.2 \pm 0.05$ | $0.1 \pm 0.03$ |
|  | DAPS | $0.05 \pm 0.02$ | $0.24 \pm 0.05$ | $0.15 \pm 0.04$ |
|  | MCGDIFF | $0.1 \pm 0.03$ | $0.34 \pm 0.06$ | $0.15 \pm 0.04$ |
| TWEEDIE | DDSMC-0.0 | $0.07 \pm 0.02$ | $0.26 \pm 0.05$ | $0.27 \pm 0.07$ |
|  | DDSMC-0.5 | $0.07 \pm 0.02$ | $0.27 \pm 0.05$ | $0.2 \pm 0.05$ |
|  | DDSMC-1.0 | $0.05 \pm 0.01$ | $0.24 \pm 0.04$ | $0.14 \pm 0.03$ |
| ODE | DDSMC-0.0 | $0.05 \pm 0.01$ | $0.23 \pm 0.05$ | $0.21 \pm 0.06$ |
|  | DDSMC-0.5 | $0.05 \pm 0.01$ | $0.23 \pm 0.05$ | $0.15 \pm 0.05$ |
|  | DDSMC-1.0 | $0.08 \pm 0.03$ | $0.4 \pm 0.05$ | $0.36 \pm 0.08$ |

*Table 10.* PSNR results on FFHQ experiments. The noise level is $\sigma_y = 0.05$, and the tasks are inpainting a box in the **middle**, outpainting right **half** of the image, and **s**uper-**r**esolution $4\times$. Numbers are averages and standard deviations over 1k images. DDRM-20 and DDRM-300 means DDRM with 20 and 300 steps respectively. **Higher is better.**

|  |  | MIDDLE | HALF | SR4 |
|---|---|---|---|---|
|  | DDRM-20 | $29.14 \pm 2.16$ | $16.19 \pm 2.51$ | $28.47 \pm 2.0$ |
|  | DDRM-300 | $30.68 \pm 2.37$ | $16.85 \pm 3.21$ | $29.14 \pm 2.1$ |
|  | DCPS | $29.74 \pm 2.01$ | $16.18 \pm 2.77$ | $26.52 \pm 1.94$ |
|  | DAPS | $29.66 \pm 2.06$ | $16.18 \pm 2.75$ | $28.28 \pm 1.88$ |
|  | MCGDIFF | $27.49 \pm 1.96$ | $14.39 \pm 2.26$ | $27.76 \pm 2.02$ |
| TWEEDIE | DDSMC-0.0 | $30.09 \pm 2.07$ | $15.86 \pm 3.1$ | $27.28 \pm 1.85$ |
|  | DDSMC-0.5 | $29.18 \pm 2.15$ | $15.78 \pm 2.98$ | $27.07 \pm 1.69$ |
|  | DDSMC-1.0 | $27.75 \pm 2.35$ | $14.46 \pm 2.39$ | $26.59 \pm 1.97$ |
| ODE | DDSMC-0.0 | $29.68 \pm 2.1$ | $15.12 \pm 2.96$ | $27.51 \pm 1.84$ |
|  | DDSMC-0.5 | $28.85 \pm 2.16$ | $15.41 \pm 2.99$ | $26.81 \pm 1.86$ |
|  | DDSMC-1.0 | $25.47 \pm 2.27$ | $11.87 \pm 1.64$ | $21.41 \pm 1.9$ |

### G.2.5. ADDITIONAL QUALITATIVE RESULTS

We provide additional results on the image reconstruction task in Figures 12 to 14.

*Table 11.* Attribution information for the FFHQ images displayed as ground truth in this paper. Images have been preprocessed (e.g., aligned and cropped) in the original curation of the FFHQ dataset. We then downsampled these to $256 \times 256$ for the experiments, and further downsampled for this paper. More details can be found in the metadata of the FFHQ dataset: `https://github.com/NVlabs/ffhq-dataset`

| IMAGE ID | AUTHOR | ORIGINAL URL | LICENSE URL |
|---|---|---|---|
| 100 | DEBBIE GALANT | `HTTPS://WWW.FLICKR.COM/PHOTOS/DEBGALANT/25236485093/` | `HTTPS://CREATIVECOMMONS.ORG/LICENSES/BY/2.0/` |
| 102 | TENACIOUSME | `HTTPS://WWW.FLICKR.COM/PHOTOS/TENACIOUSME/4674489614/` | `HTTPS://CREATIVECOMMONS.ORG/LICENSES/BY/2.0/` |
| 103 | ALPHALAB STARTUP AC-CELERATOR | `HTTPS://WWW.FLICKR.COM/PHOTOS/ALPHALAB/8880216552/` | `HTTPS://CREATIVECOMMONS.ORG/LICENSES/BY/2.0/` |
| 104 | SALESIANO SAN JOSÉ | `HTTPS://WWW.FLICKR.COM/PHOTOS/97073574@N04/9813413673/` | `HTTPS://CREATIVECOMMONS.ORG/LICENSES/BY/2.0/` |
| 110 | CAPTURED BY JOLENE C.P. HOANG | `HTTPS://WWW.FLICKR.COM/PHOTOS/HOANGJOLENE/38813884045/` | `HTTPS://CREATIVECOMMONS.ORG/LICENSES/BY/2.0/` |
| 111 | ARCHIDIECEZJA KRAKOWSKA BIURO PRASOWE | `HTTPS://WWW.FLICKR.COM/PHOTOS/ARCHIDIECEZJAKRAKOW/40877014205/` | `HTTPS://CREATIVECOMMONS.ORG/PUBLICDOMAIN/MARK/1.0/` |
| 126 | GHOLZER | `HTTPS://WWW.FLICKR.COM/PHOTOS/GEORGHOLZER/4612924700/` | `HTTPS://CREATIVECOMMONS.ORG/LICENSES/BY-NC/2.0/` |
| 135 | RESOLUTESUPPORTMEDIA | `HTTPS://WWW.FLICKR.COM/PHOTOS/ISAFMEDIA/4833372671/` | `HTTPS://CREATIVECOMMONS.ORG/LICENSES/BY/2.0/` |
| 137 | BERENDS PRODUCTIES | `HTTPS://WWW.FLICKR.COM/PHOTOS/118258384@N07/39503180405/` | `HTTPS://CREATIVECOMMONS.ORG/LICENSES/BY-NC/2.0/` |
| 146 | ELI SAGOR | `HTTPS://WWW.FLICKR.COM/PHOTOS/ESAGOR/6109130164/` | `HTTPS://CREATIVECOMMONS.ORG/LICENSES/BY-NC/2.0/` |
| 147 | INNOTECH SUMMIT | `HTTPS://WWW.FLICKR.COM/PHOTOS/115363358@N03/18076371188/` | `HTTPS://CREATIVECOMMONS.ORG/LICENSES/BY/2.0/` |
| 151 | HAMNER_FOTOS | `HTTPS://WWW.FLICKR.COM/PHOTOS/JONATHAN_HAMNER/3765961686/` | `HTTPS://CREATIVECOMMONS.ORG/LICENSES/BY/2.0/` |
| 171 | JO CHOU | `HTTPS://WWW.FLICKR.COM/PHOTOS/GIDGETS/34616317141/` | `HTTPS://CREATIVECOMMONS.ORG/LICENSES/BY-NC/2.0/` |
| 185 | OSSEOUS | `HTTPS://WWW.FLICKR.COM/PHOTOS/OSSEOUS/37453437426/` | `HTTPS://CREATIVECOMMONS.ORG/LICENSES/BY/2.0/` |
| 189 | PHIL WHITEHOUSE | `HTTPS://WWW.FLICKR.COM/PHOTOS/PHILLIECASABLANCA/3698708497/` | `HTTPS://CREATIVECOMMONS.ORG/LICENSES/BY/2.0/` |
| 190 | UNIVERSITY OF THE FRASER VALLEY | `HTTPS://WWW.FLICKR.COM/PHOTOS/UFV/26175131313/` | `HTTPS://CREATIVECOMMONS.ORG/LICENSES/BY/2.0/` |
| 199 | AMISOM PUBLIC INFOR-MATION | `HTTPS://WWW.FLICKR.COM/PHOTOS/AU_UNISTPHOTOSTREAM/40272370250/` | `HTTPS://CREATIVECOMMONS.ORG/PUBLICDOMAIN/ZERO/1.0/` |

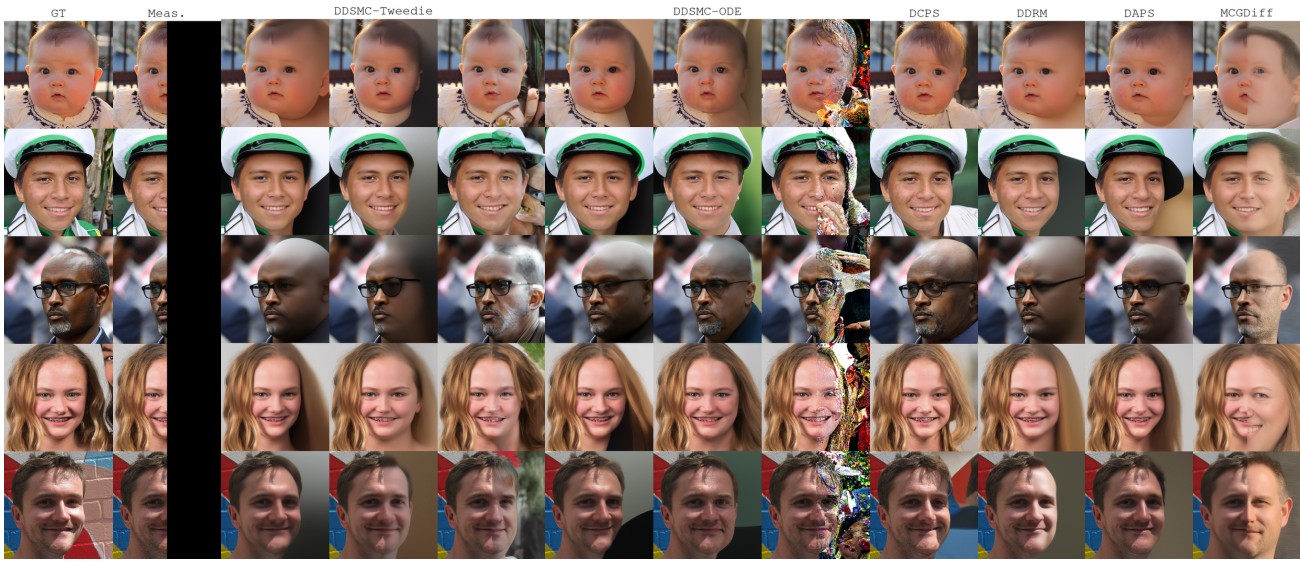

*Figure 12.* Additional results on the outpainting task. The DDSMC samples are ordered with $\eta = 0$ to the left, $\eta = 0.5$ in the middle, and $\eta = 1$ to the right. Ground truth images are FFHQ image IDs 102, 104, 199, 137, and 189 (see Table 11 for attribution).

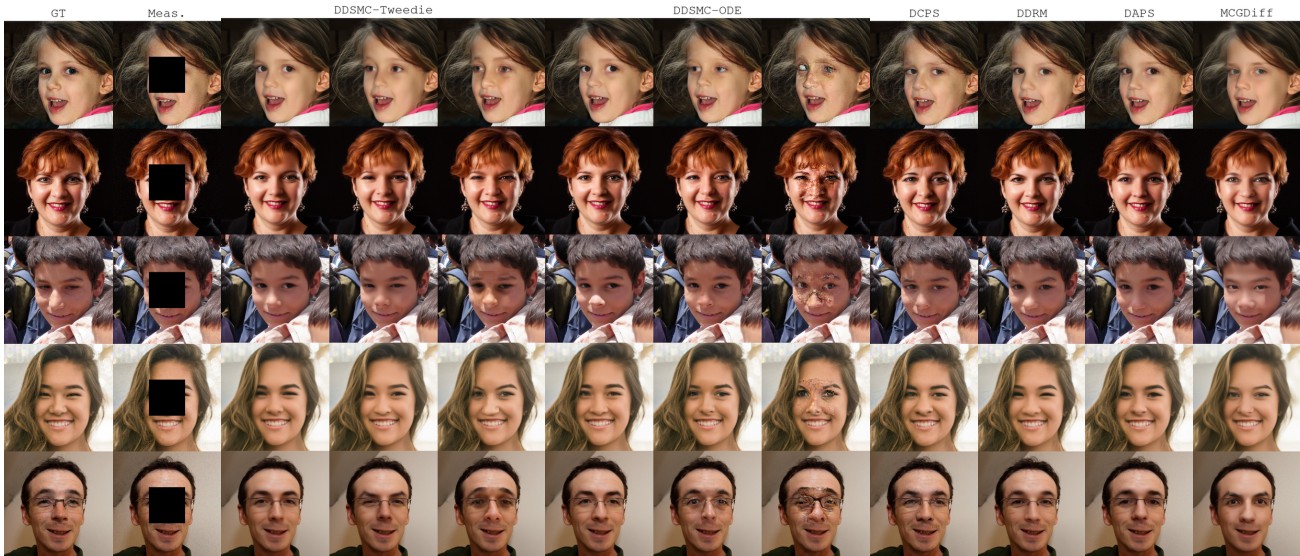

*Figure 13.* Additional results on the inpainting task. The DDSMC samples are ordered with $\eta = 0$ to the left, $\eta = 0.5$ in the middle, and $\eta = 1$ to the right. Ground truth images are FFHQ image IDs 151, 171, 135, 110, and 126, (see Table 11 for attribution).

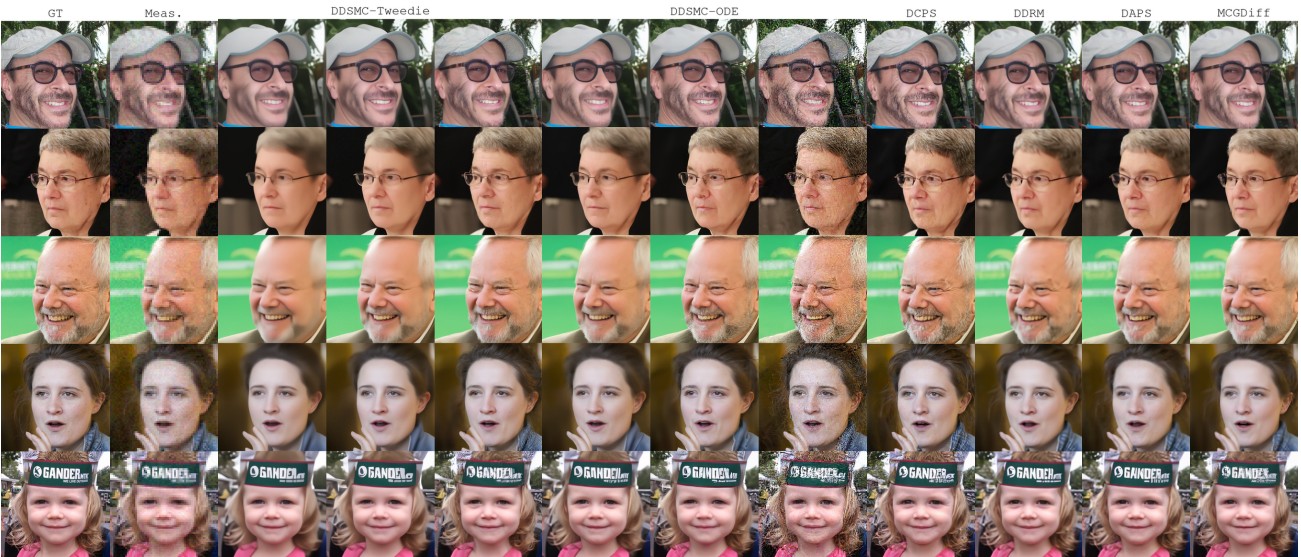

*Figure 14.* Additional results on the super-resolution ($4\times$) task. The DDSMC samples are ordered with $\eta = 0$ to the left, $\eta = 0.5$ in the middle, and $\eta = 1$ to the right. Ground truth images are FFHQ image IDs 185, 111, 190, 147, 146 (see Table 11 for attribution).

### G.3. Protein Structure Completion

#### G.3.1. IMPLEMENTATION DETAILS

We implement DDSMC in the ADP-3D codebase[18], and used DDSMC-Tweedie with $\eta = 0$. We otherwise follow the ADP-3D experimental setup, but also extend the experiments to include non-zero noise levels. For ADP-3D, we had to lower the learning rate for the higher noise levels, using 0.1 and 0.001 for $\sigma = 0.1$ and $\sigma = 0.5$, respectively. We run the algorithms with 8 different seeds. To somewhat compensate for the additional computations when running DDSMC, we use 200 steps instead of 1 000 that are used in ADP3D. As the diffusion process in $\mathbf{z}$ follows the DDPM/VP formulation, we use the same choice of $\rho_t$ as in the GMM setting.

#### G.3.2. EXTENDED RESULTS

In Figure 15, we present results on four additional proteins with PDB identifiers 7pzt (Dix et al., 2022), 7r5b (Warstat et al., 2023; Huegle, 2023), 7roa (Cruz et al., 2022; Stogios et al., 2022), and 8em5 (Cuthbert et al., 2024; Cuthbert & Goulding, 2023). In general, we see the same trend that DDSMC is competitive with ADP-3D, which specifically was designed for this problem, and that the SMC aspect with parallel particles and resampling improves performance.

---

[18] https://github.com/axel-levy/axlevy-adp-3d

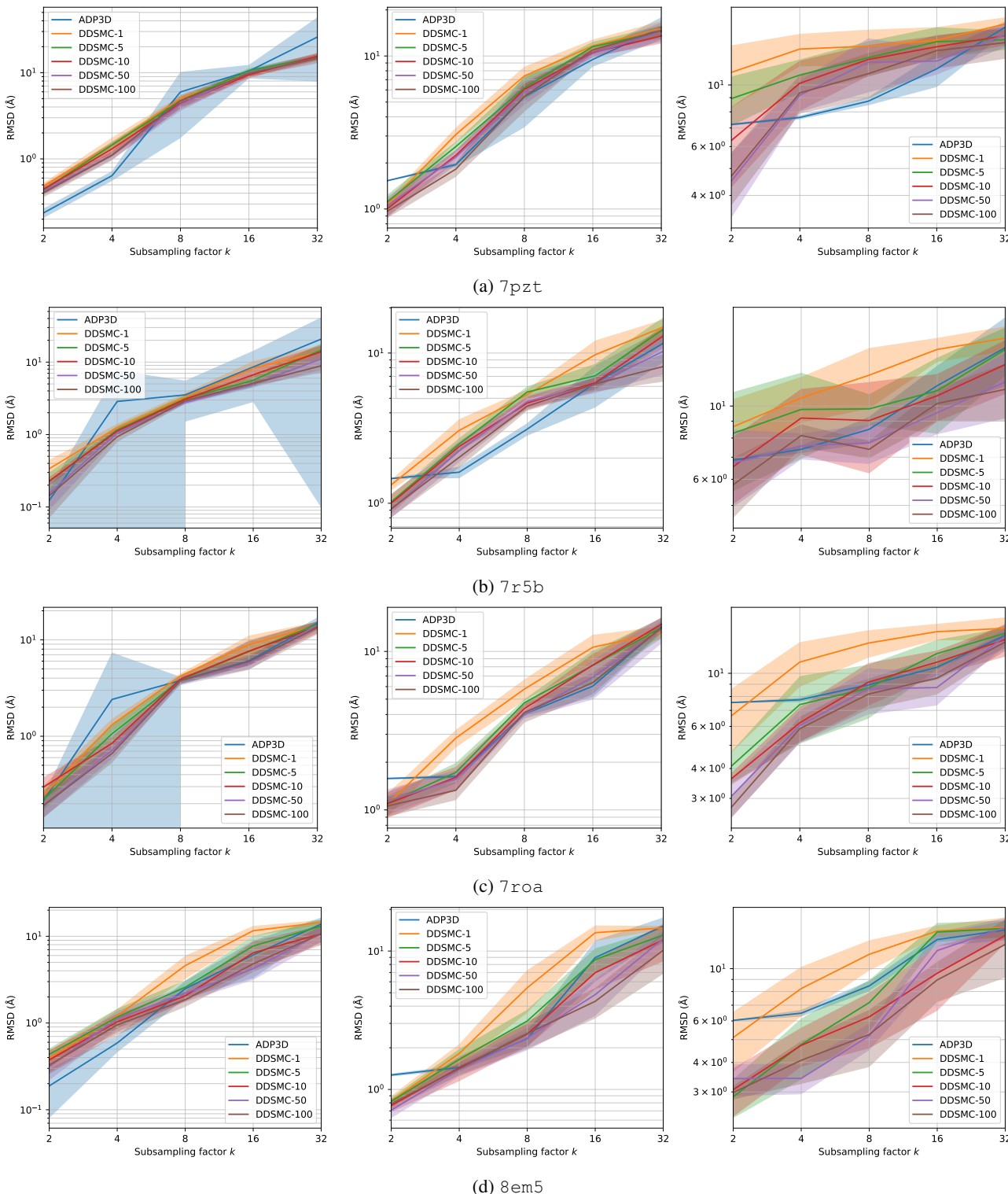

*Figure 15.* Additional results on the protein structure completion problem in Section 5.3. RMSD vs subsampling factor $k$ for additional protein with different noise levels $\sigma$: 0 (left), 0.1 (middle), 0.5 (right). Comparing ADP-3D (Levy et al., 2024) and DDSMC with different number of particles.

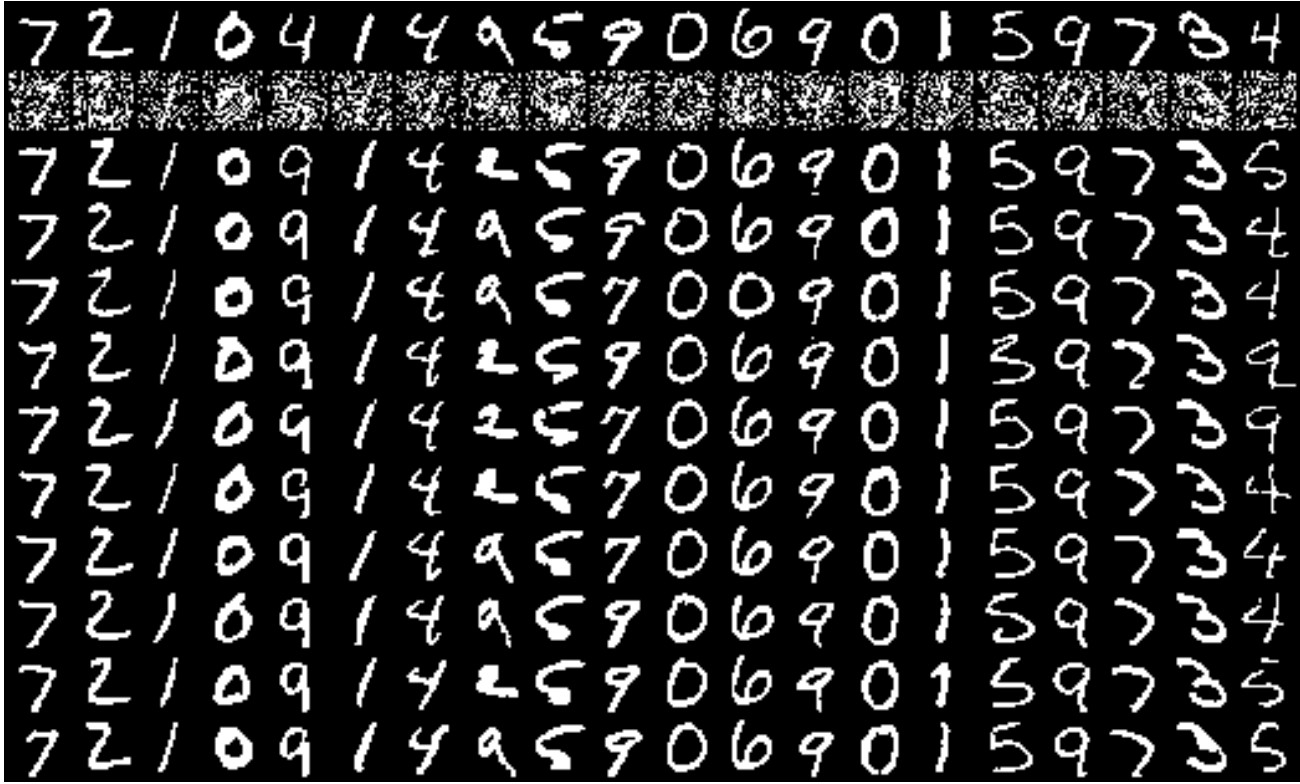

*Figure 16.* Extended qualitative results on binary MNIST using D3SMC with $N = 5$ particles. Top row is ground truth, second row the measurement, and the bottom 10 rows are samples from different independent SMC chains.

### G.4. Binary MNIST

In Figure 16 and 17 we provide extended results when running D3SMC on binary MNIST, using the same measurement model as in the main text and $N = 5$ or $N = 100$ particles, respectively.

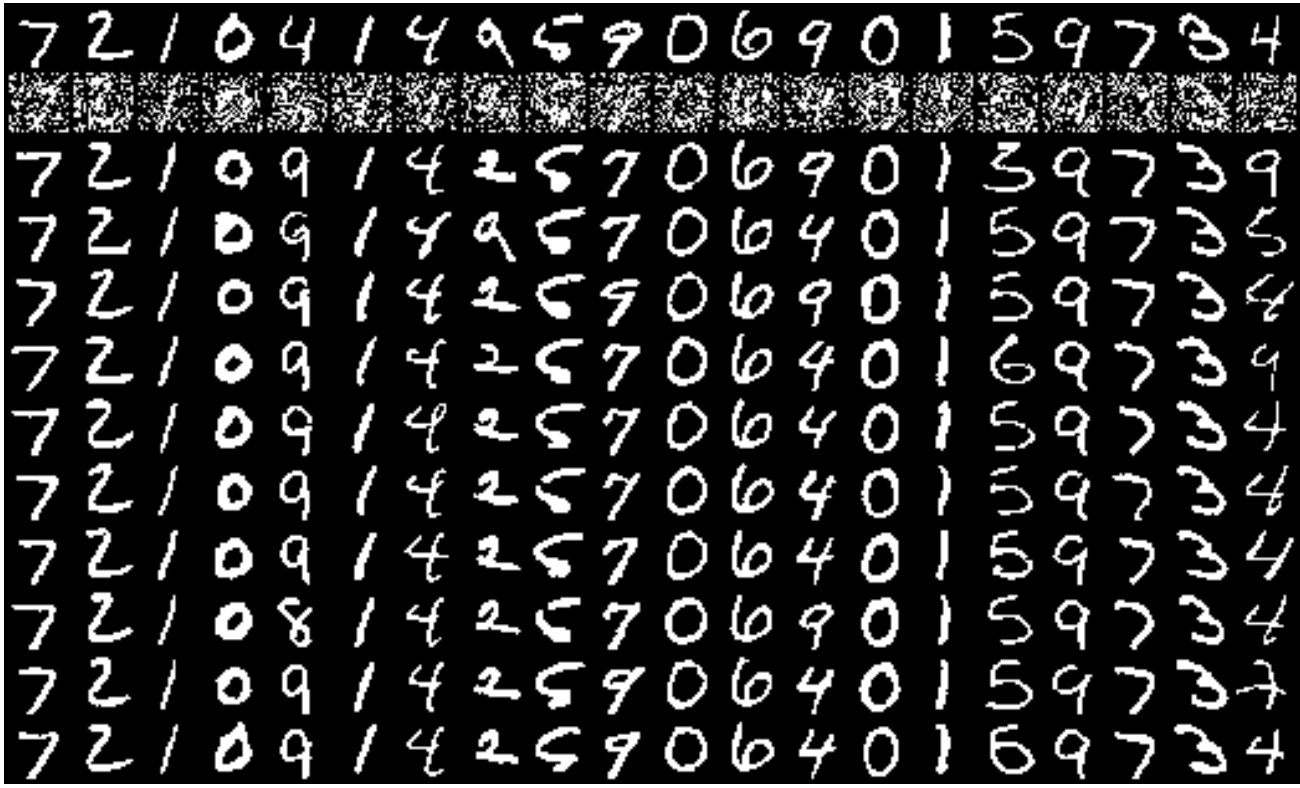

*Figure 17.* Extended qualitative results on binary MNIST using D3SMC with $N = 100$ particles. Top row is ground truth, second row the measurement, and the bottom 10 rows are samples from different independent SMC chains.

