# OpenReview forum: "Solving Linear-Gaussian Bayesian Inverse Problems with Decoupled Diffusion Sequential Monte Carlo"
_ICML.cc/2025/Conference — ICML 2025 poster_

### Official Review · Reviewer_zGWq · 2025-03-05

**Overall Recommendation:** 2

**Summary:**

## Summary
* This paper is based on previous work of solving diffusion inverse problems using sequential monte carlo [Practical and Asymptotically Exact Conditional Sampling in Diffusion Models]. More specifically, it takes the inner loop part of decoupled posterior sampling [Improving Diffusion Inverse Problem Solving with Decoupled Noise Annealing], and blend it with SMC based posteior sampling using an annealing parameter. By adjusting annealing parameter $\eta$, this new prior can generalize previous SMC based methods.
* The authors verify the effectiveness of their approach, on GMM and image restoration tasks. The proposed approach seems to be sota on GMM, and achieves competitive performance on image restoration tasks.

**Claims And Evidence:**

## Claims And Evidence
* I have some concerns with using the solution of PF-ODE as a approximated sample for q(x0|xt+1). This is approximation is not well justified, as PF-ODE is marginal preserving not distribution preserving. The PF-ODE, starting from random xt+1, has same marginal distribution as q(x0). However, whether it serves as a good approximated sample in q(x0|xt+1) is not sure.
* I am a little bit confused by the motivation of this papar. It seems that TDS [Practical and Asymptotically Exact Conditional Sampling in Diffusion Models] is already asymptotically exact. DDSMC use approximated sample such as tweedie and PF-ODE, to q(x0|xt+1). Does the approximation error harms the asymptotically exactness? Is there any theoretical advantage of DDSMC, over TDS?

**Essential References Not Discussed:**

## Essential References Not Discussed
I find no essential reference missing.

**Experimental Designs Or Analyses:**

## Experimental Designs Or Analyses
* See Methods And Evaluation Criteria

**Methods And Evaluation Criteria:**

## Methods And Evaluation Criteria
* The evaluation of this paper is a kind of limited. The authors only verify their approach on 100 FFHQ images. This limited data makes it hard to compute divergence based metrics such as FID. A relatively larger, more diverse dataset such as 1000 ImageNet images, might strengthen the empirical results.
* Further, only LPIPS is chosen as benchmark, while a more comprehensive comparsion using PSNR and FID can help readers understand the results better.
* The complexity of the proposed approach is quite high, while I find no metrics on this, such as wall clock time or FLOPS. It seems to me that DDRM can be super-fast, and DAPS can also be made fast by adjusting parameters. The Tweedie version of DDSMC can be as efficient as DAPS. However, I am really not sure about whether it is fair to compare PF-ODE DDSMC, as it appears a lot slower than DDRM and DAPS.

**Other Comments Or Suggestions:**

N/A

**Other Strengths And Weaknesses:**

N/A

**Questions For Authors:**

N/A

**Relation To Broader Scientific Literature:**

## Relation To Broader Scientific Literature
* This paper contributes a new algorithm to the diffusion inverse solvers. The main contribution is a more effective SMC based algorithm with the idea taken from DAPS.

**Theoretical Claims:**

## Theoretical Claims
* The theoretical claims look correct to me.

---

> ### Author Rebuttal · Authors · 2025-03-31
>
> Thank you for the comments, which have been of great value to improve the paper.
> ## Design choices in SMC make big difference in practice, motivating us to construct a new and better algorithm
> The reviewer correctly points out that TDS and MCGDiff already enjoy asymptotic exactness. However, the choice in intermediate targets and in proposal can make a big difference in practice with finite number of particles. Motivated by previous works on SMC for diffusion priors, we set out to design a new and improved algorithm. The different SMC algorithms indeed show very different performance in the experiments, with DDSMC outperforming both MCGDiff and TDS, providing strong evidence that the design choices made in DDSMC give better practical efficiency compared to the other SMC methods. As a reply to _"By adjusting annealing parameter $\eta$, this new prior can generalize previous SMC based methods"_ we would like to emphasize that DDSMC is a novel SMC method for the problem under study _regardless of the annealing parameter $\eta$_. In fact, our main contribution is the development of this novel SMC method, and we view the generalization of the DAPS prior (i.e., introducing $\eta$) as a secondary contribution. See also the reply to psuz. See response to tiGx and XuMA regarding asymptotic exactness
> ## We will rephrase the part about PF-ODE for sampling from $q(x_0|x_{t+1})$
> In the background section (line 104 col 1) we wrote that it is possible to use PF-ODE to sample from $q(x_0|x_{t+1})$, in order to generate a sample trajectory from the prior. We thank the reviewer for pointing out this error, and we agree that this is incorrect
>
> Note that **this paragraph was only included as an "intuitive explanation" of how the proposed method works, and the validity of the method does not in any way rely on the PF-ODE sampling from $q(x_0|x_{t+1})$**.
> What we actually meant to say with this section was that, conceptually, a (convoluted) way to simulate from the _prior_ backward process would be:
> Initialize $x_T\sim q(x_T)$.
> For $t=T-1,\dots,0$,
> 1. Solve $\hat x_{0,t+1}=\text{PF-ODE}(x_{t+1})$ from time $t+1$ to 0
> 2. Sample $x_{t}\sim q(x_{t}|x_0=\hat x_{0,t+1})$
>
> This would result in samples such that, marginally for any $t$, $x_t\sim q(x_t)$, but we do not get samples from the _joint_ $q(x_{0:T})$, _nor from the conditionals_ $q(x_0|x_{t+1})$ as the reviewer correctly points out. This sampling process motivates the DAPS prior that we use (and generalize), which is why we mentioned it in the background, but we will of course make sure to update the text so that it is mathematically correct when revising the manuscript.
> ## We have evaluated DDSMC on protein structure completion
> The reviewer writes _"The evaluation of this paper is a kind of limited."_ In the response to XuMA we have added results for another experiment concerning protein structure completion, showing that DDSMC can outperform the tailored APD-3D method out-of-the-box in (realistic) high-noise setting.
> ## We have now evaluated using 1k images
> We reran the experiments on 1k images, with essentially identical results (see response to psuz). We also computed PSNR, where now DDRM is the overall strongest model. However, just as for LPIPS, the standard deviation is rather large. Given that our method aims to recover posterior distributions, PSNR as a per-pixel metric (even stricter than per-sample metric like LPIPS) does not represent an ideal metric here, and for generative models there is often a trade-off between perceptual (e.g., LPIPS) and distortion (e.g., PSNR) metrics [1]. We will supply the PSNR table in the appendix with a comment in the main paper. We don't compute FID as we are focused on sampling from 1k different conditional distributions and not 1 unconditional distribution.
>
> [1] Blau and Michaeli, The Perception-Distortion Tradeoff, CVPR 2018
> ## Clarification regarding complexity
> We agree that a discussion is missing and will add this in the paper. In summary: DDSMC-Tweedie requires N times more (N=number of particles) NFEs per diffusion step compared with DDRM, and DDSMC-ODE has N times the NFE of DAPS. DDSMC-Tweedie has the same complexity as MCGDiff and requires slightly fewer NFEs than TDS (which requires differentiating through the score-function). See response to XuMA for an additional study using fewer particles to obtain the same number of NFEs for DDSMC-ODE as MCGDiff in the GMM case. For images, we are already using fewer NFEs as we are using fewer particles.
>
> The additional NFEs required for SMC should be viewed as way of trading off improved sample quality with compute. As seen in the additional GMM experiments with fewer particles, this aspect holds empirically as the performance improves when using more particles (especially over using a single particle). For methods like DDRM or DCPS, we have attempted to use as much computation as reasonably possible and they still fail while DDSMC effectively enjoys the compute-quality trade-off.

---

> > ### Comment · Reviewer_zGWq · 2025-04-07
> >
> > My concerns about the experimental results remain.
> >
> > The authors claim that they have reran evaluated using 1k images in the response to __psuz__. While I searched in the response to __psuz__ but find no additional result. This is a little bit confusing.
> >
> > __AC and other reviewers: have I missed any additional results?__
> >
> > The authors also have not justify the adoptation of PF-ODE in approximation of posterior sample, which is one of two major way in reconstruction of $x_0$ in their method. I think this issue is important.

---

> > > ### Author Response · Authors · 2025-04-07
> > >
> > > Thank you for your comment.
> > >
> > > First of all, we had introduced a typo, and the 1k results are discussed in the response to **tiGx**. We are sorry for the confusion. This can be found under the headline "We have now evaluated on 1k images, computed standard deviations", and reads "We took the reviewer's advice and evaluated on a 1k image validation set. The numbers, however, differ only by a maximum 0.01. The standard deviations of the 1k values ranges between 0.01 up to 0.075." In other words, we do not see much difference when using more images (the LPIPS values are the same, differing by 0.01 at maximum).
> > >
> > > ## Regarding evaluation
> > > We want to highlight that we have also made an extensive study in a Gaussian mixture model (GMM) setting, where we can really evaluate the posterior sampling capabilities, which is the task we are targeting in our paper. See the heading "We target posterior sampling, which is verified in the GMM task'' in our response to psuz. As we say there, this is a necessary study to show that the model can actually sample from the true posterior, and in the GMM setting, we can check this. Additionally, as far as we are aware, there is no way of doing this exactly for images, and therefore the main purpose of the image experiments is to first qualitatively evaluate that the model also works in high-dimensional, real-world, settings, and also quantitatively, using the LPIPS metric. As mentioned, "we are in line with SOTA methods, and outperform MCGDiff, which is the _closest comparable_ method to DDSMC"
> > >
> > > ## Regarding PD-ODE
> > > We are unsure about the new comment about using PF-ODE solution as the reconstruction. We interpreted the initial review as referring to the paragraph in the background section which stated that PF-ODE is a sample from $q(x_0|x_{t+1})$. We agree that this statement was incorrect, and in the rebuttal we described how this paragraph was intended to give an "intuitive explanation" of how the proposed method works, and how we will change that part to be mathematically correct. As we stress in the rebuttal, **the validity of our method does not rely on this paragraph**.
> > >
> > > If the new question is not about this paragraph, but in general asks about a motivation why we can use PF-ODE as the reconstruction, we would like to first highlight that this is motivated in the main paper in the paragraph describing the DAPS prior, line 134 col 1, and is also reflected in our responses to XuMA regarding Proposition A.1 and asymptotical exactness. Essentially, in the DAPS prior, we can use the PF-ODE to obtain a sample from $p_\theta(x_0)$ which can then be pushed forward in time, and this will, under some assumptions, lead to samples from the same marginal $p_\theta(x_t)$.
> > >
> > > If we still haven't answered the question, we kindly ask the reviewer for further clarifications.

---

### Official Review · Reviewer_XuMA · 2025-03-10

**Overall Recommendation:** 3

**Summary:**

The paper proposes a new SMC method for sampling from the posterior of a Bayesian inverse problem that uses the as prior the time zero marginal of a learned score-based (or diffusion) generative model. The proposed SMC is influenced by the [1] but restricts itself to a Gaussian linear likelihood and instead of Langevin proposes an SMC approach. The paper also has links with [2] and [3], which are other SMCs methods in the literature. The paper also proposes an extension to discrete diffusion which is an unique feature with respect to the other available SMC samplers.

The proposed algorithm is evaluated both in a toy dataset where a tractable Bayesian posterior distribution is available and also on image datasets. While in the toy example it excels w.r.t the other available methods, in the image datasets it comes as a second best w.r.t [4], even though the authors rightly point out that the available metrics for the image task do not exactly measure the "correctness" of the posterior sampler, but rather the visual qualities of the images.


[1] Zhang, B., Chu, W., Berner, J., Meng, C., Anandkumar,
A., and Song, Y. Improving Diffusion Inverse Problem
Solving with Decoupled Noise Annealing, July 2024
[2] Wu, L., Trippe, B., Naesseth, C., Blei, D., and Cunningham,
J. P. Practical and Asymptotically Exact Conditional
Sampling in Diffusion Models. Advances in Neural Infor-
mation Processing Systems, 36:31372–31403, December
2023.
[3] Cardoso, G., el Idrissi, Y. J., Corff, S. L., and Moulines,
E. Monte Carlo guided Denoising Diffusion models for
Bayesian linear inverse problems. In The Twelfth Interna-
tional Conference on Learning Representations, 2024.
[4] Janati, Y., Moufad, B., Durmus, A. O., Moulines, E., and
Olsson, J. Divide-and-Conquer Posterior Sampling for
Denoising Diffusion priors. In The Thirty-eighth Annual
Conference on Neural Information Processing Systems,
November 2024.

**Claims And Evidence:**

The claims of the paper concerning their performance are supported by clear and convincing evidence.
However, I feel that there is a slight problem with one of the theoretical claims, namely the validity of the SMC sampler under general conditions. Notably, in the text, the authors claim :

"SMCDiff (Trippe et al., 2023) and FPS (Dou & Song, 2023)
are two other SMC algorithms that target posterior sampling
with diffusion priors, but these rely on the assumption that
the learned backward process is an exact reversal of the
forward process, and are therefore not consistent in general."

which leads the readers to believe that this is not the case of the current approach. But I have doubts over such claim (see theoretical claims).

**Essential References Not Discussed:**

no

**Experimental Designs Or Analyses:**

I checked the soundness of the experimental designs or analysis, but I have one issue with the current analysis is that the current analysis is not made under a same-budget criterion. Indeed, for example in the mixture of gaussians example, the authors state that they used 256 particles for all SMC samplers. The problem is that this would generate a much much higher NFE (Neural function evaluation) for their algorithm, as they need to solve the whole ODE for each time and each particle, which is not the case of MCGDiff or TDS. Therefore, instead of doing 256x20 NFE as MCG diff and TDS (TDS actually does a bit more) they do approximately 20 times more.. I would suggest increasing the number of particles in MCGDiff and TDS to have a fair comparison in such example.

This concern however, does not concern the image section where an almost equivalent budget is used for MCGDIff and the proposed method.

**Methods And Evaluation Criteria:**

Yes, the benchmarks are well chosen and make sense for the application, even though I feel that an addition of a different source of real data would greatly enhance the evaluation of the current method (such as audio, or video or as in DCPS the ECG).

**Other Comments Or Suggestions:**

Line 171 second column there is an extra parenthesis in the Gaussian.

**Other Strengths And Weaknesses:**

The paper is clearly written and does a clear review of the existing methods in SMC.

Besides the two points raised above (theoretical claims and methodology), I feel however that one item that is lacking for practitioners is an analysis of the parameter sensitivity. While the authors show how the parameter $eta$ influences the performance, it is not clear how one should chose either the number of particles and the number of steps in the ODE. It would be interesting to see the tradeoffs between them in a fixed budget regime.

**Questions For Authors:**

My main question is asked in the theoretical claims section, namely:
1. Does the SMC holds under the hypothesis that the two joints distributions (backward and forward) do not match? My understanding is that this is not the case.
2. How do all the algorithms compare with equal NFE in the mixture of gaussian case?

**Relation To Broader Scientific Literature:**

Yes, the related material and the context of the proposed method are clearly explained. The paper also clearly explain the differences between the different SMC samplers.

**Theoretical Claims:**

Yes, I have checked the theoretical claims and I have an issue with proposition A.1. While the proof is correct, the usage in equation (4) is not correct. Indeed, if we assume that $f_{\theta}(x_{t+1})$ is a sample of $p_{\theta}(x_{0})$, the to obtain a sample of $p_{\theta}(x_t)$, proposition A.1 suggests that one has to use $p_{t|0}(x_t | x_{0})$ which is not equal $q_{t|0}(x_t | x_0)$ except if the forward and backward match, if I'm not mistaken. Thus, to the SMC proposed to be valid, it needs two assumptions: The ODE samples from $p_{0}(x_0)$ and that the "forward of the backward" $p_{t|0}$ is equal to the forward of the forward $q_{t|0}$.

While this is not a problem per-se and can be considered as an approximation to render the SMC tractable, the SMC is not assymptotically exact in general conditions as is the case of MCGDiff and TDS. Indeed, it would fall under the category of " SMCDiff (Trippe et al., 2023) and FPS (Dou & Song, 2023)
are two other SMC algorithms that target posterior sampling
with diffusion priors, but these rely on the assumption that
the learned backward process is an exact reversal of the
forward process, and are therefore not consistent in general."

---

> ### Author Rebuttal · Authors · 2025-03-31
>
> Thank you for taking the time to read and comment on our paper, which certainly has been useful to make the paper better. We answer concerns and questions below.
> ## Assumptions in proposition A.1 concerns DAPS vs standard prior, not asymptotical exactness of DDSMC
> Thanks for pointing out the unclarity regarding the assumptions in Proposition A.1., and how they affect the consistency of DDSMC. We emphasize that **Proposition A.1 only concerns properties of the DAPS prior**, not the DDSMC algorithm per se. Specifically, it refers to whether the DAPS prior ($\eta =0$) and the standard diffusion prior ($\eta=1$) result in the same (marginal) prior $p_\theta(x_0)$. If the assumptions do not hold (as they will not in practice), these priors are different. However, regardless of the choice of prior (we view this as a design choice, more about this below),  **DDSMC will have asymptotic exactness guarantees**, i.e., the empirical approximation will converge to the corresponding posterior *induced by the chosen prior*. It is in this last point where SMCDiff and FPS are different: they also start from a diffusion-model prior which (combined with the given likelihood) induce a posterior. However, these algorithms then target _an approximation of the induced posterior_. This approximation, which is the target of their respective SMC samplers, will correspond to the actual induced posterior only when the forward and backward kernels match. This is what we mean by "SMCDiff and FPS [...] are therefore not consistent in general".
>
> As mentioned above, it's important to note that the prior in DDSMC is a design choice. We have developed the method based on a generalization of the DAPS prior, because the decoupling offered by this prior has proven to be useful in prior work. However, the DDSMC method is equally applicable to the "standard diffusion prior" (simply set $\eta=1$) and the algorithm will then provide consistent approximations of the posterior induced by this prior. In this case we do not rely on Proposition A.1 at all.
>
> We thus believe that the assumptions in Proposition A.1 are used in a fundamentally different way in DDSMC than in SMCDiff and FPS. We thank the reviewer for highlighting this important detail, and we will make a clarification around line 144 col 1 in a revised version of the paper. See also response to reviewer tiGx about asymptotical exactness. If there are any more questions or concerns about any of this, we are happy to answer in a follow-up comment.
> ## We have tried GMM experiments with same compute budget as for MCGDiff
> Using Tweedie's formula as the reconstruction requires just a single evaluation of the score function, meaning in **this case we are already using the same compute budget as MCGDiff in the GMM experiments**. We will clarify that this is what we mean in line 358 col 1 when saying that "DDSMC outperforms all other methods, even using Tweedie's reconstruction".
>
> We do agree, though, that using DDSMC-ODE under same compute budget is also interesting, and have therefore performed additional experiments using DDSMC-ODE with 12 particles (20x less than the other methods) and 25 particles (10x less, as the number of steps in the ODE decreases from 20 to 1, i.e. $\sim 10$x more score evaluations/particle on average), and the results show that in low dimensions, 12 or 25 particles is still better than the Tweedie (and hence, MCGDiff/TDS), but this changes for higher dimensions, where using Tweedie with more particles seems to be the better choice. This shows how the SMC aspect (multiple particles and resampling) indeed is an important aspect in the performance. We will add these results in the appendix, along with a similar experiment where we change the number of ODE-steps instead, and a comment in the main paper.
> ## Additional empirical results
> We agree with the reviewer that "an addition of a different source of real data would greatly enhance the evaluation", and therefore looked at the protein structure completion in ADP-3D [1]. Their method is built for that type of task, and our method performs well out of the box, outperforming ADP-3D on higher, but realistic, noise levels (where we had to tweak their learning rate to give reasonable results).
>
> RMSD on 7qum protein. Columns indicate that every $n$ residues are observed.
> | Model ($\sigma=0$) | 2 |4 | 16 | 32 | 64 |
> |---| ---| ---| ---| ---| ---|
> | ADP-3D | 0.229|0.378 | 1.690 | 3.590| 7.788|
> | DDSMC | 0.231|0.938 | 2.385| 3.858| 8.552| 13.643 |
>
> | Model ($\sigma=0.1$) | 2 |4 | 16 | 32 | 64 |
> |---| ---| ---| ---| ---| ---|
> | ADP-3D | 1.371|1.429|3.404|4.540|8.542| 13.318|
> | DDSMC | 1.264|1.568 |2.849| 4.201| 8.927| 13.456 |
>
> | Model ($\sigma=0.5$) | 2 |4 | 16 | 32 | 64 |
> |---| ---| ---| ---| ---| ---|
> | ADP-3D | 6.704| 7.087| 7.970| 9.283| 14.441|14.387|
> | DDSMC | 6.047 | 6.245| 6.742|7.479| 10.282| 14.79178619|
>
> [1] Levy et al. Solving Inverse Problems in Protein Space Using Diffusion-Based Priors, arXiv, 2024

---

### Official Review · Reviewer_tiGx · 2025-03-11

**Overall Recommendation:** 3

**Summary:**

The authors consider solving linear inverse problems with diffusion models, but only those where the forward model has a tractable SVD.  They build on the recent decoupled annealed posterior sampling (DAPS) method by Zhang (2024) by replacing its Langevin sampling inner-loop with a sequential Monte-Carlo (SMC) sampler.  They claim that this improves posterior sampling performance and present numerical experiments with low-dimensional synthetic GMM data and high-dimensional image data.  Their method differs from several recent works on diffusion-SMC in the details of the intermediate targets and proposal function.

## update after rebuttal

I appreciate the clarifications given in the authors' response, but I still have concerns about the readability of the paper, since a lot of effort will be required to make it clear and accessible. If you look closely at my review, there are a number of unanswered questions, and responses to other reviewers suggest that there are many typos. Regarding the existence of linear inverse problems without implementable SVDs, there are many, including motion deblurring, multi-coil MRI, computed tomography, any i.i.d. random forward operator like those popular in compressive sensing, etc. With these issues in mind, I am leaving my score as-is.

**Claims And Evidence:**

The main claim is that the proposed method is asymptotically exact (see the abstract).  I did not find the explanation convincing because many approximations are made and it's not clear whether they all guarantee asymptotic exactness.
* In line 204 col 1, it's acknowledged that (13) is an approximation of the posterior.  Is this relevant to the proposed method or does it pertain only to DAPS?
* In (16) a proposal distribution is constructed based on several approximations.  Does they interfere with asymptotic exactness?
* Around (68) we learn that several key hyper parameters like $\rho_t$ are heuristically adjusted. How does the choice of $\rho_t$ affect asymptotic exactness?

Another big issue with the paper is that the proposed DDSMC method is never clearly described.  Algorithm 1 doesn't capture what is described in the text.
* It's not clear how $f_\theta$ on line 171 col 1 is computed.  Based on line 321, this seems to require an inner loop with a DDIM ODE solver, but this is not clearly described.  For example, (66) seems to describe an inner loop, but the time variable $t$ is the same as used in the outer loop of Algorithm 1, making it impossible to understand.
* Throughout the paper, the word "steps" is used in an ambiguous and confusing way.  Based on Algorithm 1, there seems to be outer "steps" but also inner steps used when evaluating $f_\theta$, but often the authors don't distinguish between them.  And there is a confusing comment in line 323 about "remaining steps $t,t-1,...,0$".  What does this mean?
* In section F.2.1, the $\sigma_t$ and $t_i$ quantities are described over $M$ steps, but these quantities do not appear in Algorithm 1, nor even the DDIM ODE update (66).  How does $M$ relate to $T$ in Algorithm 1?  How are the $t_i$ in F.2.1 related to $t$ in Algorithm 1?

**Essential References Not Discussed:**

There are various other MCMC approaches to posterior sampling with diffusion that should be discussed.  For example
* Florentin Coeurdoux, Nicolas Dobigeon, and Pierre Chainais. Plug-and-play split Gibbs sampler: Embedding deep generative priors in Bayesian inference. IEEE Trans. Image Process., 33:3496–3507, 2024.
* Zihui Wu, Yu Sun, Yifan Chen, Bingliang Zhang, Yisong Yue, and Katherine Bouman. Principled proba- bilistic imaging using diffusion models as plug-and-play priors. In Proc. Neural Info. Process. Syst. Conf., 2024.
* Xingyu Xu and Yuejie Chi. Provably robust score-based diffusion posterior sampling for plug-and-play image reconstruction. In Proc. Neural Info. Process. Syst. Conf., 2024.

**Experimental Designs Or Analyses:**

One issue with the experimental analysis is that the number of NFEs is never clearly reported for the proposed method.  It seems to grow as the product of the number of particles $N$, the number of outer loop steps $T$, and possibly the number of inner loop steps $M$, but as reported earlier, the proposed method is never clearly described.  In any case, it is imperative to explicitly list the number of NFEs, which seems to be a serious drawback of the method.

Another issue with the experiments is that the authors use only a 100-image validation set, whereas 1000 is typical in most respected diffusion posterior sampling papers.  Just because DCPS and DAPS used 100 doesn't mean that it is sufficient.  With only 100 validation images, there is likely to be large standard errors on the averaged performance metrics, and no standard errors are even presented in Table 3.

**Methods And Evaluation Criteria:**

It's not clear that the authors evaluated the competing methods under appropriate hyper parameter choices.  For example, the DDRM paper shows in their Table 6 that both PSNR and FID get worse as the NFEs are increased from 20 to 100, suggesting that "more is not better".  But in the paper under review, DDRM was evaluated with 1000 or 300 NFEs, which are massively larger than the standard value of 20, without justification.  This may be a very poor choice.

**Other Comments Or Suggestions:**

I suggest the authors pay more attention to clearly describing the proposed method and clearly stating and defending the main claims.

**Other Strengths And Weaknesses:**

I don't think the authors have been forthcoming on the computational burden of their method, as the NFEs used were never clearly stated. I suspect that the computational requirements are massive, making the method uninteresting for practical application.

Also, I don't think the authors have been forthcoming on the restriction of their method to linear inverse problems with implementable SVDs.  This is a strong restriction.

Furthermore, while the authors dismiss other non-consistent SCM works (see line 249, col 2), they have not clearly established the consistency of their approach.

**Questions For Authors:**

* In line 94 col 1, the expression for $p_\theta(x_t|x_{t+1})$ is described as an "approximation", but it seems instead like a definition, given the goal stated in the first paragraph of the Background section.  In other words, given $q$ and $f_\theta$, we define $p_\theta(x_t|x_{t+1})$ from them and then aim to sample from $p_\theta(x|y)$. Is that correct? Same question applies to (4).
* In line 076 col 2, I wonder if there is a typo in the definition of the resampling step, because it is a circular definition.

**Relation To Broader Scientific Literature:**

Personally, I think that this paper is a relatively minor variation on other recent SMC diffusion posterior sampling works like MCDiff and TDS.

**Theoretical Claims:**

It would help to have a detailed theorem and proof statement for the "asymptotically exact" claim.  Currently the claim is too vague.

---

> ### Author Rebuttal · Authors · 2025-03-31
>
> We thank the reviewer for their comments, and we address their concerns below (see answer to zGWq regarding complexity and NFEs).
>
> ## We will clarify the asymptotical exactness guarantees
> SMC provides consistent approximations of its sequence of targets $\lbrace \pi_t(x_{t:T})\rbrace_{t=0}^T$ under weak conditions. However, since we only care about the final target, it is enough that $\pi_t(x_{0:T})$ admits the true posterior $p_\theta(x_0|y)$ as a marginal for the method to be consistent. This is the case for DDSMC by construction; see Eq (7) with $t=0$. The _intermediate targets_ $(\pi_t(x_{t:T}), t>0)$ as well as the _proposals_ can be seen as design choices that affect the efficiency of the algorithm but not its consistency. The particular ways in which we design these quantities (while ensuring correctness of the _final target_) constitute the DDSMC framework.
> We mention this general fact in line 92 column 2 regarding the intermediate targets but will add a comment regarding the proposal.
> We note that the specific questions that you ask regarding approximations in Eq (13), (16), (68) _only affect the intermediate targets and/or proposal_ and thus not the consistency of the final target approximation.
>
> We realize that this should be further clarified, and we will hence add a "theorem-like" paragraph, making it clear how our algorithm will be asymptomatically exact. See also response to XuMA.
> ## We view reconstruction function $f_\theta$ as a design choice
> In our experiments, we tried two types of reconstruction functions: either using Tweedie's formula (see line 87 col 1) or the PF-ODE. As we view this as a design choice, we have not explicitly written how $f_\theta$ is computed in Algo 1. It is true, however, that if using the PF-ODE, this requires an inner loop. In eq (66) there is a typo, where $t$ should be replaced by $t'$ (the inner loop time variable), and in line 323 we mean that we start at $x_{t'} = x_t$, then use Eq 66 for $t'=t-1, t-2, \dots, 0$. I.e., we start from a sample at the diffusion time (outer loop index) $t$, then convert that into a sample $x_0$ using as many steps in the inner loop as there are "left" in the outer loop. We will fix these typos and clarify line 323.
>
> This flexibility also affects the computational cost/NFE, where the Tweedie reconstruction avoids the outer loop and is therefore more efficient. Compared to MCGDiff and TDS, DDSMC-Tweedie has the same number of NFE (but avoids the expensive differentiation of the reconstruction network in TDS), but can still provide better result. The DDSMC framework is general and choice between Tweedie and PF-ODE (or some other reconstruction method) becomes a computational trade-off for the user.
> ## While falling under the same framework as TDS and MCGDiff, we provide better empirical performance
> Both TDS and MCGDiff are SMC methods, and we differ in how we design both the proposal and targets, which empirically shows improved performance. Additionally, we make a secondary contribution in the choice of prior (using $\eta\neq 1$), which can further improve performance.
>
> ## We evaluate DDRM with fewer steps, gives worse performance
> We used 300 steps when running DDRM to match that of DCPS, but on the reviewers advice we also tried 20, which shows worse result. We will add that to the appendix. For the GMM case we used 1k steps as more steps would mean less discretization error, and instead using 20 shows worse performance for 8 and 80 dimensions, while being similar at 800 dimensions.
> ## We have now evaluated on 1k images, computed standard deviations
> We took the reviewer's advice and evaluated on a 1k image validation set. The numbers, however, differ only by a maximum 0.01. The standard deviations of the 1k values ranges between 0.01 up to 0.075.
> ## We are not aware of linear inverse problems without implementable SVDs
> We make it very clear that this method tackles linear inverse problems (it is in the title) which is still an open question (e.g., MCGDiff and DCPS tackle exactly this). However, we are not aware of linear inverse problems which do not have implementable SVDs and would gladly appreciate pointers to examples of this if implementable SVDs is a large concern.
> ## Other points
> * We will add a paragraph in Related work to cite the suggested papers and discuss related MCMC methods.
> * We agree that the expression for $p_\theta(x_t|x_{t+1})$ is a definition in the current problem formulation (although based on approximating $q(x_t|x_{t+1})$ when training the generative model) and will clarify
> * In the resampling step, a set of "ancestor indices" $`{`a_t^i`}`_{i=1}^N$ are sampled from the Multinomial distribution, and each particle $x_t^i$ is then replaced by $x_t^{a_t^i}$. The circular dependency comes from overloading the notation. We will clarify to avoid confusion
> * The notation in F.2.1. came from following the model formulation used by DAPS, which gave rise to a notational inconsistency. We will make sure to clarify

---

### Official Review · Reviewer_psuz · 2025-03-12

**Overall Recommendation:** 2

**Summary:**

The paper introduces Decoupled Diffusion Sequential Monte Carlo (DDSMC), a method for Bayesian inverse problems using diffusion priors. Main contributions include: Leveraging a modified diffusion process ("DAPS prior") to enable larger updates during sampling, improving exploration. Combining SMC with diffusion models to provide asymptotically exact posterior sampling, addressing limitations of prior methods that rely on approximations. Extending the approach to discrete data (D3SMC) via discrete diffusion models (D3PM).

## update after rebuttal

Based on the authors' rebuttals, and also the reviews of other reviewers, I would like to maintain my original rating.

**Claims And Evidence:**

N/A

**Essential References Not Discussed:**

No.

**Experimental Designs Or Analyses:**

- The contribution appears incremental when compared to existing methods such as DAPS and the experimental results do not fully demonstrate a clear advantage over current methods. Overall, the novelty and significance of the work may not be sufficient for acceptance at ICML.
 - The main limitation lies in the experiments. A large portion of the results relies on synthetic Gaussian mixture models. Since Gaussian mixture models provide ground truth score information, the results based on them do not capture the challenges encountered by existing diffusion model-based methods in real-world applications. For the real-world FFHQ dataset, experiments in inpainting, outpainting, and super-resolution yield performance of DDSMC that is not clearly superior to other methods. For example, DCPS ranks first in all tasks in Table 3, casting doubt on the practical benefits of the proposed DDSMC method.
- The proposed DDSMC method reduces to existing methods in extreme cases (e.g., inverse temperature eta = 0 and using PF-ODE for reconstruction). The paper should demonstrate how varying eta affects the results and that an intermediate value between 0 and 1 offers better performance. However, Table 1 shows that with 800 dimensions for x, eta = 0 yields the best performance, and Table 3 reveals similar results for eta = 0 and eta = 0.5. The results presented do not effectively support the claimed benefits of the inverse temperature. Additionally, the experiments omit analysis on the number of particles for SMC. Since other methods run only once, DDSMC appears to gain an unfair advantage, particularly against methods that also involve random sampling. Experiments using a single particle for DDSMC and multiple runs for existing methods are necessary.

Algorithms that merge SMC with decoupled diffusion in an innovative manner could have strengthened the paper. Using SMC to increase the number of samples for approximation and employing the inverse temperature eta might enhance performance. However, based on the experimental discussions above, the overall advantage of the proposed method appears limited. Because of this, I do not recommend acceptance.

**Methods And Evaluation Criteria:**

N/A

**Other Comments Or Suggestions:**

No.

**Other Strengths And Weaknesses:**

No.

**Questions For Authors:**

No.

**Relation To Broader Scientific Literature:**

N/A

**Theoretical Claims:**

N/A. No theoretical results in this submission (I did not check the supplementary material).

---

> ### Author Rebuttal · Authors · 2025-03-31
>
> We thank the reviewer for taking the time to comment on our paper. We have tried to address all your comments below, but there were a few points that we did not quite understand, so we kindly ask you to clarify if we in fact misunderstood some of your points.
>
> ## Merging SMC with decoupled diffusion is our core contribution
> As the reviewer comments _"algorithms that merge SMC with decoupled diffusion in an innovative manned could have strengthened the paper"_, we want to highlight that this is exactly the core contribution that we make in our paper! As far as we are aware, this is the first time that SMC is merged with decoupled diffusion, and we are therefore not sure what this comment refers to. We are happy to answer any follow up clarifications on this.
>
> The reviewer comments that _"The proposed DDSMC method reduces to existing methods in extreme cases (e.g., inverse temperature eta = 0 and using PF-ODE for reconstruction)."_  We do not agree with this claim since it misses the point that our core contribution is an **SMC algorithm** building on the (generalized) DAPS prior. Hence, this claim would only be correct if we also restrict DDSMC to using a single particle, but the "parallel particles" are a key ingredient in SMC. Running an SMC algorithm with a single particle corresponds to sampling from the proposal, which if using $\eta=0.0$ and PF-ODE would be more or less equivalent to DAPS (if instead using $\lambda_t^2=\sigma_t^2$ in eq (16)). To verify that, we ran DDSMC-ODE on the GMM case with a single particle, and the numbers obtained with $\eta=0.0$ are essentially identical to DAPS (differing by at most 0.03 from each other). Hence, we can conclude that the **introduction of the SMC framework is the key ingredient that leads to the improved performance over DAPS**. We will incorporate these results in the appendix, and make a comment in the experiment section in the main paper. Next, we do agree that a further analysis of the number of particles would be beneficial, and as part of the response to XuMA, we performed additional experiments with fewer particles using the DDSMC-ODE.
>
>
> ## We target posterior sampling, which is verified in the GMM task
> We agree that our experiments largely depend on synthetic experiments. However, we believe that this is a necessary sanity check for a method such as DDSMC which is designed to target the __correct posterior distribution__, intended to empirically prove that our proposed technique samples from the true posterior.
> In order to achieve this, having an exact score/ground truth posterior is necessary to control errors.
>
> The purpose of the image experiments is to first show *qualitatively* that our method works also in high-dimensional, real-world problems, and we included the LPIPS metric to also show this quantitatively: we are in line with SOTA methods, and outperform MCGDiff, which is the *closest comparable* method to DDSMC.
>
> We emphasize that our focus is on recovering the posterior distribution, and we do not aim to improve the per-sample quality like DDRM, but instead, the population quality. However, a problem is that most image-related tasks lack the ground truth posterior, and even a good approximation thereof. To our knowledge, there are no principled metrics to gauge such performance. As such, the synthetic experiment is the only setting we can rely on to compare the methods' posterior sampling ability in a principled way. We revise our Experiment section to reflect on this.
>
> ## We have evaluated DDSMC on protein structure determination
> The reviewer comments that _"The main limitation lies in the experiments."_ In the response to Reviewer XuMA we have added results for another experiment concerning protein structure determination. We show that DDSMC can outperform the tailored APD-3D method out-of-the-box in (realistic) high-noise scenarios.
> Please see response to XuMA for further details.
>
> ## We can see clear effects of changing the inverse temperature $\eta$
> As mentioned above, we view the DDSMC method itself as our primary contribution, and the _generalization of the DAPS prior_ (such that we can interpolate between DAPS and the standard diffusion prior using the inverse temperature $\eta$), as a secondary contribution. The reviewer comments that we should demonstrate "how varying $\eta$ affects the result and that an intermediate value between 0 and 1 offers better performance". We agree with the observation that $\eta=0.0$ is the best in high dimensions for DDSMC-ODE, but we emphasize that $\eta=0.5$ is better for DDSMC-Tweedie in this case, and is also the best choice in lower dimensions for DDSMC-ODE, while $\eta=1.0$ seems to be better for DDSMC-Tweedie in lower dimensions. It hence certainly seems to be an interplay between the dimension of the data, the choice of reconstruction function, and the choice of $\eta$. We already discuss this in line 304 col 2 and onward, but will extend the discussion in a revised version of the paper.

---

> > ### Comment · Reviewer_psuz · 2025-04-08
> >
> > Thank you for the responses. However, my primary concern persists. I still believe that, in comparison to DAPS, the proposed approach represents an incremental improvement, and the experiments are mainly synthetic and not sufficiently convincing.

---

> > > ### Author Response · Authors · 2025-04-08
> > >
> > > We thank you for your reply. We take the opportunity to once again stress that we have developed a method which targets posterior sampling, and not a designated image reconstruction method (while DAPS claim that they target posterior sampling, we show in our experiments that the introduced approximations make the model fail in doing so). As mentioned in the rebuttal, we have tried an experiment on proteins and we see that DDSMC with multiple particles consistently outperforms a single particle (which as mentioned in the rebuttal essentially is equivalent to DAPS), see tables below. We think this again (in addition to the GMM experiments) show that the introduction of the SMC aspect is an **important and non-negligible contribution** in our work.
> > >
> > > RMSD on 7qum protein, lower is better. N is the number of particles. Columns indicate that every $n$ residues are observed.
> > > | N ($\sigma=0$) | 2 |4 | 8| 16| 32 |
> > > |---| ---| ---| ---| ---| ---|
> > > | 1 | 0.339 | 1.111 | 2.683 | 7.590 | 12.010 |
> > > | 100 | 0.231|0.938 | 2.385| 3.858| 8.552| 13.643 |
> > >
> > > | N ($\sigma=0.1$) | 2 |4 | 8| 16| 32|
> > > |---| ---| ---| ---| ---| ---|
> > > | 1 | 1.297 | 1.775 |3.072| 7.897 | 13.230 |14.871 |
> > > | 100 | 1.264|1.568 |2.849| 4.201| 8.927| 13.456 |
> > >
> > > | N ($\sigma=0.5$) | 2 |4 | 8| 16| 32|
> > > |---| ---| ---| ---| ---| ---|
> > > | 1 | 6.216 | 6.695 | 7.254| 11.036 | 12.825| 15.610 |
> > > | 100 | 6.047 | 6.245| 6.742|7.479| 10.282| 14.79178619|

---

### Decision · Program_Chairs · 2025-05-01

**Decision:**

Accept (poster)

**Comment:**

The authors propose a new sequential Monte Carlo method for linear-Gaussian inverse problems using diffusion models. Following previous works on posterior sampling with diffusion models such as MCGdiff and TDS, the authors use a proposal which uses information about the observations. They also follow DAPS to introduce a proposal which can be evaluated explicitly in the linear and Gaussian setting considered in the paper. The method is evaluated empirically in a Gaussian mixture models setting and compared to various state-of-the-art samplers. An experiment on image restoration is also proposed along with a proof of concept with discrete data.

During the discussion, the authors clarified Proposition A.1 and the theoretical foundations of their method to establish asymptotic guarantees. They also clarified some misleading statements in the original paper (the PF-ODE sampling for instance) which should be included in the revised version to improve the paper.

I believe that the GMM setting is very interesting to assess the performance of the model and aligns with previous works as it introduces a potentially very challenging posterior sampling problem with known ground truth. However, some reviewers were also concerned by the lack of more challenging numerical experiments. The authors proposed an additional experiment from [1] (Levy et al. Solving Inverse Problems in Protein Space Using Diffusion-Based Priors, arXiv, 2024) where their method outperforms the baseline for high noise levels. They also evaluate the experiment on image restoration on 1k images and propose to support the new results with a PSNR table. The fact that their approach outperforms TDS and MCGDiff which are natural SMC baselines is convincing numerically. Although I do not think it is indispensable for publication, I agree that adding a paragraph on the complexity of the method would be interesting.

For all these reasons, I lean towards acceptance of the paper.

Best regards.